# Population-level morphological analysis of paired CO₂- and odor-sensing olfactory neurons in *D. melanogaster* via volume electron microscopy

**Jonathan Choy[1†‡], Shadi Charara[1†], Kalyani Cauwenberghs[1], Quintyn McKaughan[1§], Keun-Young Kim[2], Mark H Ellisman[2], Chih-Ying Su[1]\***

[1]Department of Neurobiology, University of California, San Diego, La Jolla, United States; [2]National Center for Microscopy and Imaging Research, Center for Research in Biological Systems, University of California, San Diego, La Jolla, United States

\*For correspondence:
c8su@ucsd.edu

†These authors contributed equally to this work

Present address: §MD Program, Wayne State University School of Medicine, Detroit, United States; ‡Graduate Program in Molecules, Cells and Organisms, Harvard University, Cambridge, United States

Competing interest: The authors declare that no competing interests exist.

## eLife Assessment

This **valuable** study reveals surprising morphological diversity of *Drosophila* sensory neurons. Using serial block-face electron microscopy, the authors created detailed 3D reconstructions of large neuronal populations, **convincingly** finding significant structural variation both within and across distinct classes. These results form the basis for testable hypotheses on how neuronal arborization is optimized for particular sensory functions. This research will be highly relevant to biologists in the fields of physiology, insect chemosensation, and neuroscience.

**Abstract** Dendritic morphology is a defining characteristic of neuronal subtypes. In *Drosophila*, heterotypic olfactory receptor neurons (ORNs) expressing different receptors display diverse dendritic morphologies, but whether such diversity exists among homotypic ORNs remains unclear. Using serial block-face scanning electron microscopy on cryofixed tissues, we analyzed the majority of CO₂-sensing neurons (ab1C) and their odor-sensing neighbors (ab1D) in the *Drosophila melanogaster* antenna. Surprisingly, ab1C neurons featured flattened, sheet-like dendrites—distinct from the cylindrical branches typical of odor-sensing neurons—and displayed remarkable diversity, ranging from plain sheets to tube-like structures that enclose several neighboring dendrites, forming 'dendrite-within-dendrite' structures. Similarly, ab1D dendrites varied from simple, unbranched forms to numerously branched morphologies. These findings suggest that morphological heterogeneity is common even among homotypic ORNs, potentially expanding their functional adaptability and ranges of sensory physiological properties.

## Introduction

Dendrites are neuronal processes specialized to receive information. In observing the bewildering variety of dendritic morphologies, Ramón y Cajal famously postulated that 'all the morphological features displayed by neurons appear to obey precise rules that are accompanied by useful consequences' (*Cajal, 1995*). What might these useful consequences be? Dendritic size and complexity vary to meet the functional demands of specific neurons (*Hall and Treinin, 2011*; *Jan and Jan, 2001*; *Jan and Jan, 2010*). For instance, the dendritic arborizations of somatosensory neurons define the geometry and size of their receptive fields (*Hall and Treinin, 2011*). On the other hand, the numerous or elongated dendrites of olfactory receptor neurons (ORNs) are thought to enhance sensory surface

area for heightened sensitivity (*Challis et al., 2015*; *Smith, 2008*). This is exemplified by the *Manduca* moth, whose long ORN dendrites in trichoid sensilla are correlated with the insect's exquisite pheromone acuity (*Kaissling et al., 1989*; *Kaissling, 1996*; *Keil, 1989*; *Lee and Strausfeld, 1990*).

Dendritic morphologies are thus broadly used as defining features for specific neuronal subtypes (*Jan and Jan, 2010*). Indeed, in *Drosophila melanogaster*, ORNs expressing different receptors or housed in distinct morphological types of sensilla exhibit diverse dendritic structures. For example, the outer dendrites of basiconic and intermediate ORNs are numerously branched, whereas those of coeloconic and trichoid neurons are typically unbranched (*Nava Gonzales et al., 2021*; *Shanbhag et al., 1999*). Using serial block-face scanning electron microscopy (SBEM) and nanoscale morphometric analysis, we further found that the sensory surface areas of outer dendrites, where olfactory receptors are localized (*Benton et al., 2006*; *Benton et al., 2009*), vary significantly among these morphological classes. ORNs with numerously branched dendrites generally have greater surface areas. Specifically, coeloconic neurons have the smallest sensory surface (1–9 $\mu m^2$), followed by trichoid (18–44 $\mu m^2$), intermediate (11–47 $\mu m^2$), and basiconic ORNs, which exhibit the largest range (11–275 $\mu m^2$) (*Nava Gonzales et al., 2021*). Interestingly, ORN types with extensive dendritic branches also exhibit larger, mitochondria-rich inner dendritic segments (*Nava Gonzales et al., 2021*), suggestive of high metabolic demands to sustain such structures. Therefore, diverse dendritic morphologies likely support specialized functions, with basiconic ORNs' highly branched dendrites enhancing food odor sensitivity, while coeloconic ORNs' short and unbranched dendrites reduce metabolic costs for detecting volatile acids or amines.

Does similar morphological diversity exist among homotypic ORNs expressing the same receptor? Addressing this question is crucial for understanding whether or how structural variations within a single neuronal type may contribute to fine-tuning odor sensitivity or additional specialized functions. In our published SBEM study, we reconstructed 3D models for four Or67d-expressing trichoid ORNs from the same antenna. Two neurons had unbranched dendrites, while the others had three and six dendritic branches, respectively. Interestingly, one unbranched dendrite displayed a cylindrical morphology proximally, transitioned to a slightly flattened structure mid-length, and reverted to a cylindrical form distally (*Nava Gonzales et al., 2021*). While these four neurons represent only a small subset of the 55–60 Or67d neurons in an antenna (*Grabe et al., 2016*; *Shanbhag et al., 1999*), these observations suggest that dendritic morphology among homotypic neurons may be less uniform than previously assumed. However, determining the full extent of dendritic heterogeneity within an identified ORN population requires a systematic morphological and morphometric analysis at the population level.

To this end, here we generated a new SBEM volume encompassing nearly the entire antenna of *D. melanogaster*. The tissues were processed using the CryoChem method, which we previously developed to achieve high-quality ultrastructural preservation of cryofixed samples for volume EM (*Tsang et al., 2018*). Cryofixation via high-pressure freezing and freeze substitution is particularly crucial for properly preserving tissues with cuticles (such as insect sensory appendages) that are impermeable to chemical fixatives. Additionally, cryofixation offers superior morphological preservation to then allow for more accurate morphometric measurements, unlike chemical fixation which can distort membrane structures, thus precluding faithful quantification (*Shanbhag et al., 1999*; *Shanbhag et al., 2000*; *Shanbhag et al., 2001*; *Tsang et al., 2018*).

Furthermore, through genetic labeling of multiple identified ORNs in SBEM volumes, our prior studies established that within a sensillum, the rank order of ORN sizes corresponds to the neurons' relative extracellular spike amplitudes, designated as 'A', 'B', 'C', or 'D' ORNs in descending order of spike size (*Nava Gonzales et al., 2021*; *Tsang et al., 2018*; *Zhang et al., 2019*). Therefore, once the sensillum identity is confirmed, the cellular identities of ORNs can be inferred based on their relative neuronal sizes, without requiring genetic labeling. Moreover, once the cellular identity of an ORN is established, its corresponding olfactory receptor can be determined based on published molecular and functional atlases of fly ORNs (*Benton et al., 2009*; *Benton et al., 2025*; *Couto et al., 2005*; *Fishilevich and Vosshall, 2005*; *Hallem et al., 2004*; *Task et al., 2022*; *Yao et al., 2005*).

Using these established and validated approaches, we characterized the 3D structures and measured the nanoscale morphometrics of $CO_2$-sensing ab1C neurons and their odor-sensing ab1D neighbor in *D. melanogaster*. $CO_2$ is a particularly ethologically significant odorant, as it serves as an alarm signal emitted from fruit flies (*Suh et al., 2004*). Additionally, the nanoscale features of these

neurons have not been previously characterized. Of note, ab1C and ab1D neurons reside within the ab1 sensillum, the most abundant basiconic sensillum type on the antenna (*Grabe et al., 2016*), making it an ideal model for examining homotypic neurons' morphologies at a population level.

In this study, we reconstructed 3D models for over 50% of the ab1C and ab1D neurons within a single antenna, enabling a broad-scale survey of morphological variability. Notably, ab1C neurons exhibited flattened, sheet-like dendrites, in stark contrast to the cylindrical branches typical of odor-sensing neurons. Both ab1C and ab1D dendrites displayed remarkable diversity, suggesting that morphological heterogeneity is a common feature among homotypic ORNs. Our findings suggest that while homotypic ORNs exhibit characteristic dendritic motifs, their morphologies are more variable than Cajal's 'precise rules' might imply.

## Results

The primary olfactory organ in *Drosophila* is the antenna, which houses hundreds of olfactory sensilla on the surface of its third segment (*Figure 1—figure supplement 1A*). Each sensillum typically encapsulates the outer dendrites of two to four ORNs. The outer dendrites are the sites where odorant receptors are expressed, enabling the detection of volatile chemicals. A small portion of the outer dendrites lies beneath the base of the sensillum cuticle. At the ciliary constriction, the outer dendrites connect to the inner dendritic segment, which then links to the soma of each ORN (*Figure 1—figure supplement 1B*; *Nava Gonzales et al., 2021*; *Ng et al., 2020*; *Shanbhag et al., 1999*; *Su et al., 2009*).

To survey *Drosophila* $CO_2$-sensing ab1C neurons at the population level, we first generated an SBEM volume covering the antennal region where ab1 sensilla are located, named the ab1 zone, on the proximal medial surface of the third antennal segment (*Figure 1A*, *Figure 1—figure supplement 1A*; *de Bruyne et al., 2001*; *Grabe et al., 2016*; *Shanbhag et al., 1999*). Notably, ab1 is the only antennal large basiconic sensillum type that houses four ORNs (*Shanbhag et al., 1999*), allowing us to unequivocally identify individual ab1 sensilla based on their relatively large size and distinctive set of four associated neurons without having to rely on genetic labeling.

Our survey of the antennal SBEM volume identified 39 ab1 sensilla, accounting for over 80% of the total ab1 population (48 sensilla), as previously documented (*Shanbhag et al., 1999*). Instead of being uniformly distributed within the ab1 zone, the ab1 sensilla were arranged in two clusters: a larger, more diffuse cluster on the proximal medial surface and a smaller cluster on the lateral side of the antennal surface (*Figure 1A*).

Interestingly, we observed substantial variation in the number of dendritic branches in the ab1 sensilla, ranging from 65 to 295 branches across 32 sensilla with quality EM images that allowed for clear distinction of individual dendrites (*Figure 1B and C*). These numbers were estimated by counting dendrites around the midpoint of the sensilla, yielding an average of 136±29 branches (mean ± SD, n=32, representing 67% of the ab1 population). This result aligns with a previous transmission electron microscopy (TEM) study, which reported an average of 95±12 ab1 dendritic branches from images at unspecified positions along the longitudinal sensillum axis (*Shanbhag et al., 1999*). The wide range of dendritic branch counts, reflected by a coefficient of variation (CV) of 0.22, indicates considerable heterogeneity among ab1 sensilla.

Does the heterogeneity in dendritic counts reflect variability in ab1 sensillum cuticle morphology? To address this question, we measured the length (L), surface area (SA), and volume (V) of 31 intact ab1 sensilla, excluding those with truncated structures. The ab1 cuticle morphometric features (L=9.50 ± 0.98 µm; SA = 93.75 ± 5.83 µm$^2$; V=46.09 ± 3.91 µm$^3$, n=31) were comparable to our published ab1 measurements from two independent SBEM volumes (L=10.80 ± 0.16 µm; SA = 70.70 ± 0.55 µm$^2$; V=39.49 ± 2.12 µm$^3$, n=5) (*Nava Gonzales et al., 2021*). Although these morphometric features also exhibited variation across the ab1 population, their CVs were markedly smaller (CV$_L$ = 0.10, CV$_{SA}$ = 0.06, CV$_V$ = 0.08) compared to that of dendritic branch counts (CV$_{DC}$ = 0.22) (*Figure 1C*). This analysis suggests that sensillum cuticle morphology is less variable than dendritic counts. Furthermore, sensilla with higher dendritic counts are not necessarily thicker than those with fewer branches (see *Figure 1—source data 1* for ab1 sensillum morphometrics). Indeed, an analysis of midpoint dendritic number vs sensillum cuticle cross-sectional area revealed no correlation ($R^2 \approx 0$, *Figure 1D*), suggesting that cuticular morphology and ORN dendritic number are regulated independently.

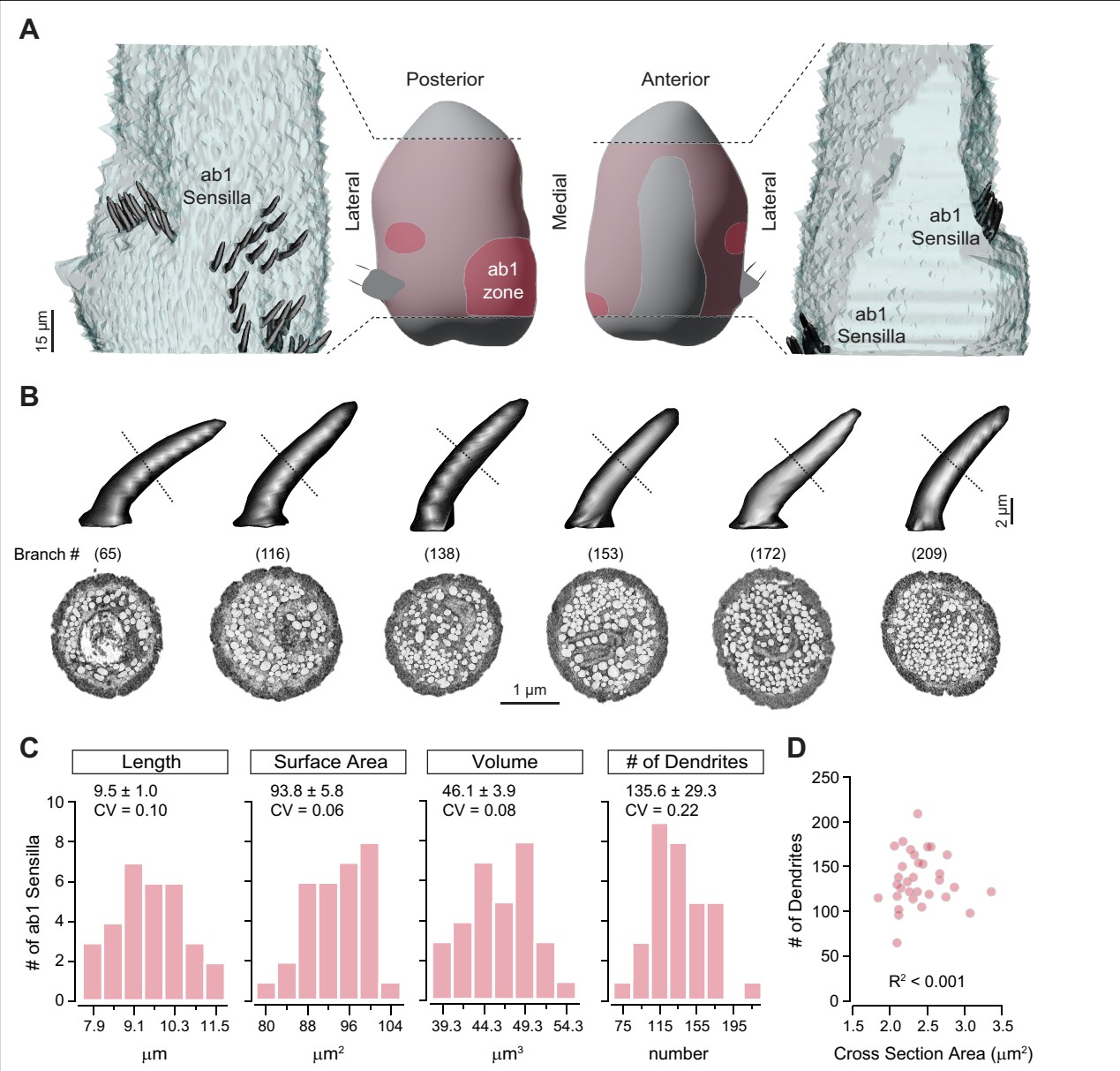

**Figure 1.** Distribution of ab1 sensilla and their morphometric analysis. (**A**) Middle panels: Illustration of the antennal region sampled in the serial block-face scanning electron microscopy (SBEM) volume (pink) with the ab1 zone highlighted in pink. Left and right panels: 3D models of ab1 sensillum cuticles (dark gray) are shown on the imaged portion of the antenna (light gray). Scale bar: 15 µm. (**B**) 3D models and corresponding SBEM images of ab1 sensilla. Dashed lines indicate the approximate midpoint region of cuticles where the SBEM images were sampled. Sensilla are arranged from left to right in order of increasing dendritic branch counts, as indicated in parentheses. Dendrites are pseudocolored in white. Scale bars: 2 µm for 3D models and 1 µm for SBEM images. (**C**) Distribution of morphometric features (length, surface area, volume, and dendritic branch counts) from fully segmented ab1 sensillum cuticles. Mean ± SD and coefficients of variation (CVs) are shown above each graph (n=31–32). (**D**) Correlation analysis of ab1 dendritic branch counts as a function of sensillum midpoint cross-sectional areas (n=32). Also see *Figure 1—source data 1* for ab1 sensillum morphometrics.

The online version of this article includes the following source data and figure supplement(s) for figure 1:

**Source data 1.** ab1 sensillum morphometrics.

**Figure supplement 1.** *Drosophila* antenna and olfactory sensillum.

## ORNs in the ab1 sensillum

The ab1 sensillum houses four ORNs (*Figure 2A*), each expressing a specific receptor or receptor complex: Or42b (ab1A), Or92a (ab1B), Gr21a/Gr63a (ab1C), and Or10 (ab1D) (*Couto et al., 2005*; *de Bruyne et al., 2001*; *Hallem et al., 2004*; *Jones et al., 2007*; *Kwon et al., 2007*). These neurons

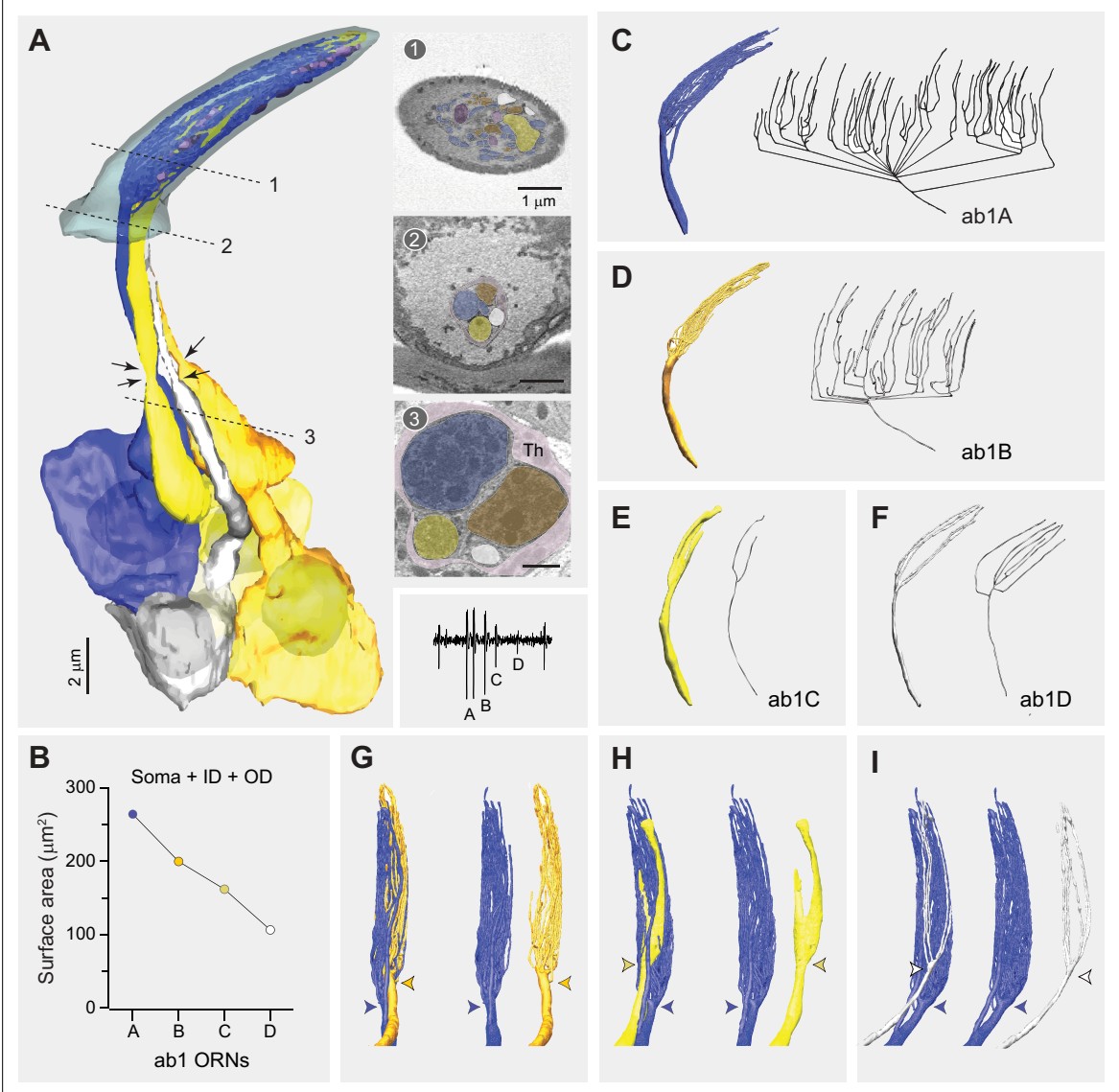

**Figure 2.** Fully reconstructed ab1 sensillum with four neurons. (**A**) 3D model and serial block-face scanning electron microscopy (SBEM) images of a fully reconstructed ab1 sensillum. Olfactory receptor neurons (ORNs) are pseudocolored to indicate neuronal identities: ab1A (blue), ab1B (orange), ab1C (yellow), and ab1D (white). Dashed lines mark the positions of corresponding SBEM images on the right: (1) sensillum lumen containing the outer dendrites; (2) proximal region of the outer dendrites; and (3) the inner dendrites surrounded by processes of the thecogen cell (Th). Arrows indicate the positions of ciliary constriction, which demarcates the inner and outer dendritic segments. Inset: a representative trace from single-sensillum recording showing the relative extracellular spike amplitudes of the ab1 ORNs. Scale bars: 2 μm for 3D models and 1 μm for SBEM images. (**B**) Combined surface areas of the ORN soma, inner dendrite (ID), and outer dendrites (OD). (**C–F**) 3D models and corresponding 2D projections of the outer dendritic branches of ab1A (**C**), ab1B (**D**), ab1C (**E**), and ab1D (**F**). (**G–I**) Spatial relationship between the dendritic branches ab1A and those of its neighboring neurons. Colored arrowheads indicate the primary branching points or flattening position. See also *Figure 2—source data 1* for ab1 ORN morphometrics and *Video 1*.

The online version of this article includes the following source data and figure supplement(s) for figure 2:

**Source data 1.** ab1 olfactory receptor neuron (ORN) morphometrics.

**Figure supplement 1.** Partially reconstructed ab1 sensillum with four neurons.

exhibit distinct extracellular spike amplitudes (*Figure 2A*), with a relative size ratio of 5.2 (ab1A):4.5 (ab1B):2.3 (ab1C):1 (ab1D) (*Zhang et al., 2019*). Our published studies have shown that, within a sensillum, the rank order of ORN sizes reflects their relative extracellular spike amplitudes (*Nava Gonzales et al., 2021*; *Tsang et al., 2018*; *Zhang et al., 2019*). This allowed us to assign the cellular identities of ab1 ORNs based on their relative neuronal sizes without having to rely on genetic labeling.

We successfully segmented all four ORNs from an ab1 sensillum (*Figure 2A*). As expected, the co-housed ORNs differed in size, and based on their sizes in descending order, we assigned neuronal identities as ab1A, B, C, and D (*Figure 2B*). Similar to the large-spike neurons in the other two large basiconic sensilla (ab2A and ab3A), both ab1A and ab1B displayed numerous dendritic branches, with total counts of 67 and 37, respectively. Given that the segmented ab1 sensillum had the fewest total dendritic branches, it is likely that other ab1A and ab1B neurons exhibit more numerous dendritic branches. As with other basiconic ORNs, the branching patterns of ab1A and ab1B were complex, featuring multiple branching points. From the base of the sensillum cuticle to its tip, the primary branches of ab1A or ab1B neurons bifurcated multiple times, generating secondary branches that further divide into tertiary, quaternary, or higher-order branches (*Figure 2C and D*). This complex branching pattern explains why the midpoint dendritic count of this sensillum (65 branches) was lower than the combined total of 104 branches for its ab1A and ab1B.

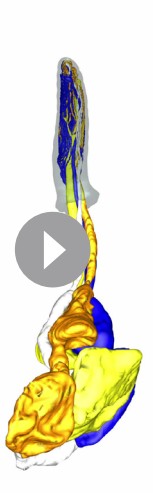

**Video 1.** 3D dendritic models of ab1A, B, C, and D neurons. Color codes are as indicated in *Figure 2*. https://elifesciences.org/articles/106389/figures#video1

In contrast, the neighboring ab1C and ab1D neurons exhibited simpler branching patterns, characterized by fewer branches and a lower degree of branching (ab1C: 2 and ab1D: 7 branches; *Figure 2E and F*). Notably, the outer dendrite of the $CO_2$-sensing ab1C had a flattened, sheet-like morphology in its distal region (*Figure 2E*), contrasting with the narrow, cylindrical branches of its odor-sensing neighbors (*Figure 2C, D, and F*). Interestingly, this dendritic sheet appeared to split into two (*Figure 2E*), distinct from the intact ab1C dendritic sheet reconstructed from another SBEM volume (*Figure 2—figure supplement 1*). This observation suggests the possibility of morphological diversity among ab1C dendrites, a question we addressed in detail in the following section.

The outer (sensory) dendrites, starting at the ciliary constriction, comprised two distinct segments: an unbranched cylindrical segment located below the sensillum cuticle base and a branched (or flattened) segment situated within the sensillum lumen (*Figure 2A*). Of note, the outer dendrite of ab1A began branching more proximally near the cuticle base compared to the neighboring neurons (*Figure 2G–I*), unlike other types of co-housed basiconic ORNs, whose dendrites typically start branching at a similar level near the sensillum cuticle base (*Nava Gonzales et al., 2021*). Furthermore, the dendritic branches of ab1A appeared to envelop those of the neighboring neurons, occupying the circumference region of the sensillum lumen (*Video 1*). We observed similar dendritic branching features in another ab1 sensillum partially sampled from a previously generated antennal SBEM volume (*Figure 2—figure supplement 1*), suggesting that the described morphologies are representative of ab1 ORNs.

## Morphological diversity across ab1C outer dendrites

To investigate the potential diversity in ab1C dendritic morphologies, we segmented the $CO_2$-sensing neurons from all ab1 sensilla with high-quality images for 3D reconstruction. Indeed, our results revealed remarkable morphological heterogeneity among individual ab1C neurons. Specifically, dendritic flattening occurred at positions ranging from 13% to 53% of the cuticle length above the sensillum base, with an average position of 40%, while dendritic termination varied between 62% and 98% of the cuticle length, averaging 89%. In general, dendrites that began flattening closer to the base tended to terminate earlier before reaching the cuticle tip (*Figure 3A*). Furthermore, these diverse dendritic morphologies can be categorized into four types (*Figure 3A*, inset) as detailed below (also see *Video 2*).

### Loosely curled category

The majority of ab1C dendritic sheets fell into the 'loosely curled' category, where the longitudinal edges of the flattened segment gently curved inward in a single intact structure. In EM cross-sections, these dendritic sheets typically exhibited a 'C'-shaped profile (*Figure 3B*, Images 1 and 2). Some cross-sections resembled an inverted 'U', as if the dendritic sheet had folded along the longitudinal axis (*Figure 3B*, Image 3); or an 'S' profile when the two edges curled in opposite directions (*Figure 3B*, Image 4). In one instance, a central ridge extended along the sheet's longitudinal axis to form a raised protrusion on the otherwise flat dendritic structure (*Figure 3B*, Image 5).

### Fully curled category

Surprisingly, a sizeable minority of ab1C dendritic sheets formed completely curled, ring-shaped, or tube-like structures that encircled several dendritic branches from neighboring neurons, forming a 'dendrite-within-dendrite' structure (*Figure 3C*). Within this category, the cross-sectional profiles of dendritic sheets varied from a completely enclosed ring (*Figure 3C*, Image 1) to a lasso-like shape (*Figure 3C*, Image 2), and even a more intricate 'θ'-like configuration (*Figure 3C*, Image 3). Interestingly, similar ring-shaped or complex dendritic profiles were previously observed in TEM images of a large basiconic sensillum containing four neurons, although the neuronal identity was undetermined (*Shanbhag et al., 1999*).

### Split category

Another subset of ab1C dendritic sheets split into two or more distinct parts (*Figure 3D*, also see *Figure 2E*). Within this category, the dendrites also exhibited remarkable morphological heterogeneity. For example, one dendrite split longitudinally into two parts, with one further bifurcating at the distal region (*Figure 3D*, Image 1). Another example displayed an uneven division, with a smaller rectangular sheetlet and a larger one resembling a loosely curled structure (*Figure 3*, Image 2). A third example demonstrated a complex morphology, with both sheetlets curling into a 'U' shape in opposite directions (*Figure 3D*, Image 3).

### Mixed category

The fourth morphological type exhibited the most intricate structure and was classified as the 'mixed' category. These dendritic sheets combined features of both the 'fully curled' and 'split' categories (*Figure 3E*). In one instance, the dendritic sheet divided into two parts: one formed a fully enclosed tubular structure, while the other sheetlet wrapped around this tube (*Figure 3E*, Image 1). In another example, one dendritic sheetlet formed a fully enclosed, blind-ended tube at the distal end, whereas the other terminated prematurely with a slight bifurcation, extending only about one-third the length of the first sheetlet (*Figure 3E*, Image 2).

## ab1C morphometrics

How might dendritic flattening influence sensory function? To address this question, we compared the morphometric features of the cylindrical and flattened segments of ab1C outer dendrites (*Figure 4*, left panel, segments 1 and 2, respectively). The cylindrical segment extended from the ciliary constriction below the sensillum cuticle base to approximately 40% of the cuticle length above the base, where dendritic flattening occurred (*Figure 3A*). On average, the flattened segment was half as long as the cylindrical segment, with average lengths of 5.74 µm and 10.72 µm, respectively. Despite its shorter length, the surface area of the flattened segment was about 22% larger than the cylindrical counterpart. In contrast, the volume of the flattened segment was much smaller (1.03 µm³), only about 40% of the cylindrical counterpart (2.64 µm³, *Figure 4A* and *Table 1*).

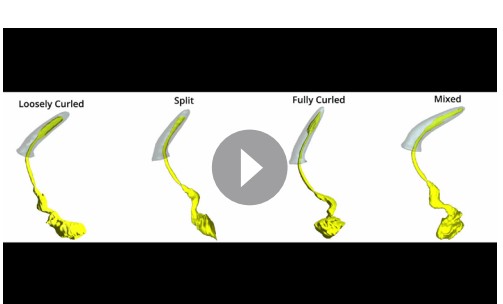

**Video 2.** 3D models representing distinct morphological types of ab1C dendrites.
https://elifesciences.org/articles/106389/figures#video2

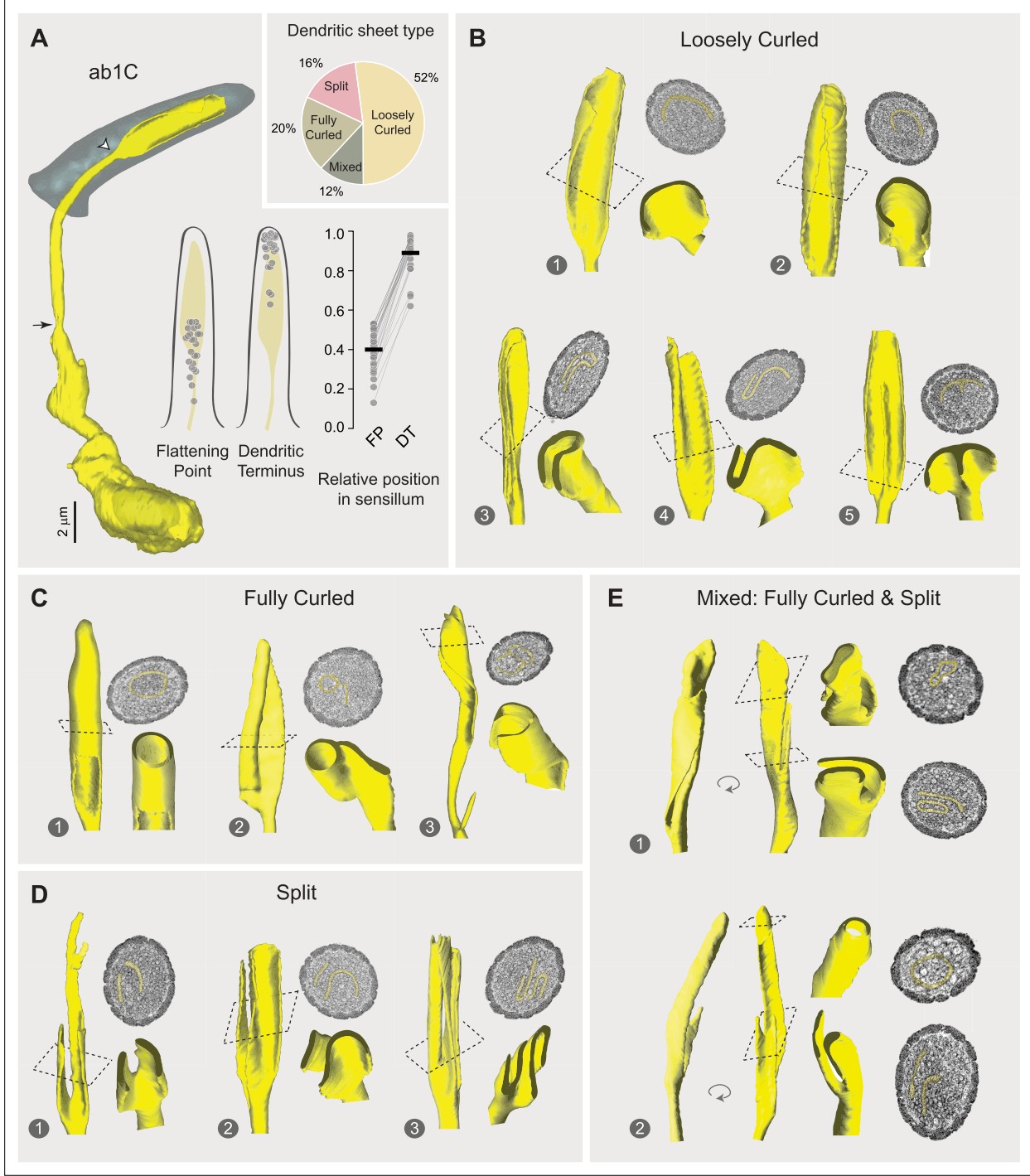

**Figure 3.** Morphological diversity across ab1C outer dendrites. (**A**) Left panel: 3D model of an ab1C neuron (yellow) and its associated sensillum cuticle (gray). Arrow indicates ciliary constriction, marking the beginning of the outer (sensory) dendrite. White arrowhead marks the location where dendritic flattening occurs. Middle panel: Positions of the flattening point (FP) and dendritic terminus (DT) relative to the cuticle length. Filled gray circles represent the relative positions of individual ab1C neurons. Right panel: Similar to the middle panel, but with lines connecting data points from the same neurons. The cuticle base and tip are designated as 0 and 1, respectively. Horizontal bars indicate the mean positions (n=25). Inset: Pie chart illustrating the distribution of the four outer dendritic morphological categories across the ab1C population. (**B–E**) Representative 3D models, top-down clipped views, and corresponding cross-sectional EM images are shown for each of the four morphological categories: loosely curled (**B**), fully curled (**C**), split (**D**), and mixed (**E**). ab1C dendritic sheets are pseudocolored in yellow in sample EM images. See also *Video 2*.

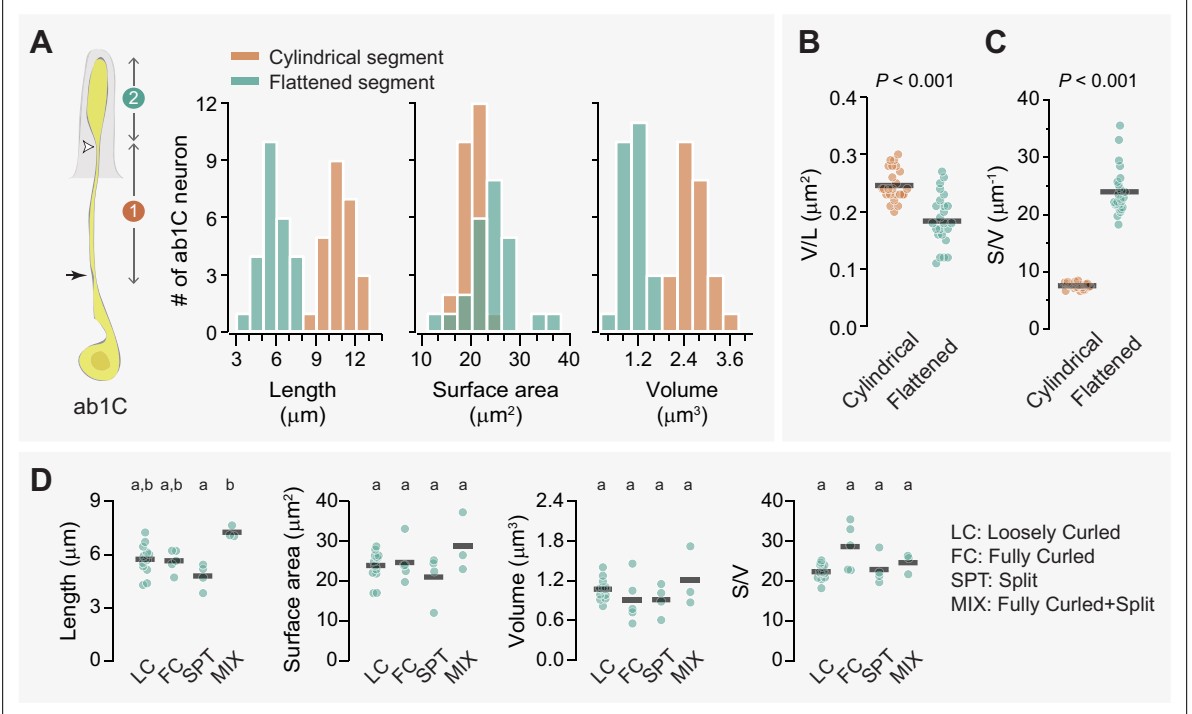

**Figure 4.** Flattening increases the surface area-to-volume ratio of the ab1C outer dendrite. (**A**) Left panel: Illustration of an ab1C neuron. Arrow indicates the position of ciliary constriction. White arrowhead marks the location where dendritic flattening occurs. Segment #1: cylindrical outer dendrite; segment #2: flattened outer dendrite. Right panels: Distributions of length, surface area, and volume for the cylindrical (coral) or flattened (teal) outer dendritic segments across the ab1C population (n=25). (**B–C**) Quantitative comparisons of the volume per unit length (**B**) and surface area-to-volume ratio (**C**) for the cylindrical (coral) and flattened (teal) outer dendritic segments. Each filled circle represents data from an individual neuron, with horizontal bars indicating the mean values (n=25). Statistical significance was determined using paired two-tailed t-test. (**D**) Quantitative comparisons of morphometric features for ab1C flattened segments across different morphological categories. Each filled circle represents data from an individual ab1C neuron, with horizontal bars indicating the mean values. Statistical significance is determined by Kruskal-Wallis one-way ANOVA on ranks and denoted by different letters. For example, labels 'a' and 'b' indicate a significant difference between groups (p<0.05), whereas labels with identical or shared letters (e.g., 'a' and 'a', 'a,b' and 'a', or 'a,b' and 'b') indicate no significant difference. Also see *Figure 4—source data 1* for ab1C dendrite morphometrics.

The online version of this article includes the following source data for figure 4:

**Source data 1.** ab1C dendrite morphometrics.

These morphometric comparisons suggest that dendritic flattening increases the surface area-to-volume ratio. Indeed, the average volume per unit length (or cross-sectional area) of the flattened segment was about 28% smaller than that of the cylindrical segment (0.18 vs 0.25 $\mu m^2$, *Figure 4B*, and *Table 1*), indicating modest tapering of the dendrite toward its tip. Strikingly, the surface area-to-volume ratio of the flattened segment was over 300% as high as the cylindrical segment (23.94 vs 7.58, *Figure 4C*, and *Table 1*). This could potentially enhance the dendrite's capacity to detect $CO_2$ and facilitate signal propagation as the increased surface area allows for more efficient passive spread of electrical signals along the dendrite. Notably, none of the morphometric features differ significantly among ab1C dendrites in different morphological categories (*Figure 4D*), implying that the morphological diversity may have evolved to serve additional functions. Overall, the marked increase in the surface area-to-volume ratio underscores the potential functional advantages of dendritic flattening in sensory processes.

## Heterogeneity in ab1D branching patterns

For comparison, we analyzed the neighboring odor-sensing ab1D, selected for its relatively low number of dendritic branches, making it suitable for systematic analysis. As previously noted, ab1D was identified based on its smallest soma and inner dendrite among the four ab1 ORNs (see *Figure 2B*

**Table 1.** Morphometric data for the outer dendritic segments of ab1C and ab1D (mean ± SD).

| ORN type | Length (µm) | | | Surface area (µm²) | | | Volume (µm3) | | | V/L | | | S/V | | |
|---|---|---|---|---|---|---|---|---|---|---|---|---|---|---|---|
| **ab1C** | Proximal | Distal | Total | Proximal | Distal | Total | Proximal | Distal | Total | Proximal | Distal | Total | Proximal | Distal | Total |
| Loosely curled (n=13) | 10.67±0.86 | 5.73±0.84 | 16.40±1.41 | 19.82±2.10 | 23.90±3.69 | 43.72±3.91 | 2.60±0.34 | 1.07±0.16 | 3.68±0.36 | 0.24±0.02 | 0.19±0.04 | 0.22±0.02 | 7.64±0.44 | 22.32±1.96 | 11.93±0.93 |
| Fully curled (n=5) | 11.62±0.88 | 5.64±0.62 | 17.27±0.78 | 20.86±2.03 | 24.66±5.03 | 45.52±4.37 | 2.73±0.26 | 0.91±0.35 | 3.64±0.43 | 0.24±0.03 | 0.16±0.06 | 0.21±0.03 | 7.67±0.72 | 28.61±5.80 | 12.56±1.07 |
| Split (n=4) | 10.76±0.59 | 4.78±0.71 | 15.54±0.48 | 20.33±2.61 | 21.00±6.12 | 41.33±7.32 | 2.80±0.49 | 0.91±0.23 | 3.71±0.66 | 0.26±0.03 | 0.20±0.07 | 0.24±0.05 | 7.32±0.48 | 22.88±3.82 | 11.21±1.60 |
| Mixed (n=3) | 9.38±0.68 | 7.26±0.33 | 16.64±0.93 | 18.07±1.43 | 28.92±7.44 | 47.00±6.73 | 2.40±0.16 | 1.21±0.45 | 3.61±0.30 | 0.26±0.03 | 0.17±0.07 | 0.19±0.02 | 7.54±0.66 | 24.58±2.51 | 12.99±0.84 |
| Total (n=25) | 10.72±0.97 | 5.74±0.95 | 16.47±1.19 | 19.90±2.09 | 24.19±4.89 | 44.09±4.81 | 2.64±0.33 | 1.03±0.25 | 3.67±0.39 | 0.25±0.03 | 0.18±0.05 | 0.22±0.03 | 7.58±0.50 | 23.94±3.93 | 12.07±1.11 |
| **ab1D** | | | | | | | | | | | | | | | |
| Unbranched (n=8) | NA | NA | 17.47±1.85 | NA | NA | 13.29±1.18 | NA | NA | 0.69±0.08 | NA | NA | 0.040±0.006 | NA | NA | 19.50±2.13 |
| Branched (n=13) | 8.94±1.11 | 29.09±10.25 | 38.83±10.05 | 8.97±1.72 | 10.77±3.30 | 19.74±4.12 | 0.60±0.10 | 0.30±0.10 | 0.90±0.20 | 0.068±0.014 | 0.010±0.003 | 0.025±0.006 | 15.03±1.63 | 37.51±4.18 | 22.25±2.65 |
| Total (n=21) | | | 30.20±12.78 | | | 17.28±4.57 | | | 0.82±0.18 | | | 0.030±0.010 | | | 21.20±2.81 |

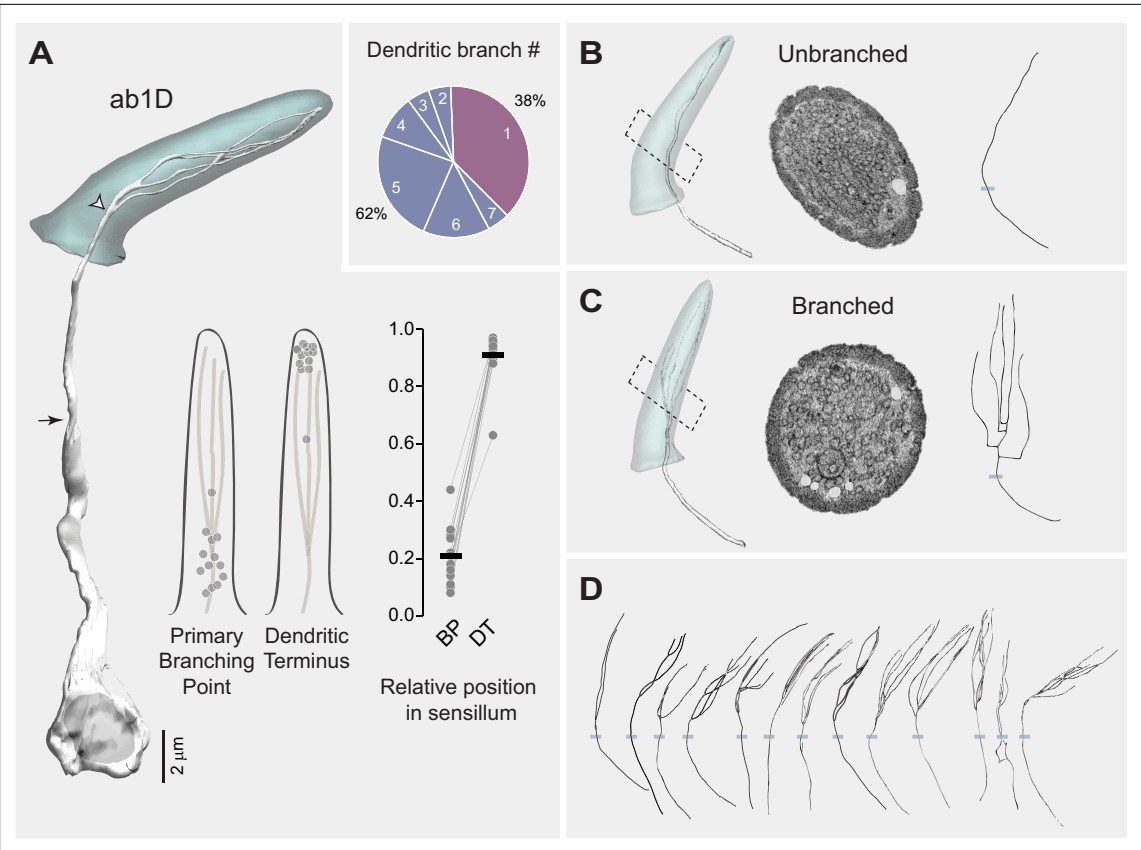

**Figure 5.** Heterogeneity in ab1D branching pattern. (**A**) Left panel: 3D model of an ab1D neuron (white) and its associated sensillum cuticle (gray). Arrow indicates ciliary constriction, marking the beginning of the outer (sensory) dendrite. White arrowhead marks the primary branching point where dendritic branching starts. Middle panel: Positions of the primary branching point (BP) and dendritic terminus (DT) relative to the cuticle length. Filled gray circles represent the relative positions of individual ab1D neurons with branched dendrites. Right panel: Similar to the middle panel, but with lines connecting data points from the same neurons. The cuticle base and tip are designated as 0 and 1, respectively. Horizontal bars indicate the average positions of the segmented ab1D neurons (n=13). Inset: Pie chart illustrating the distribution of the two dendritic branching categories across the ab1D population (n=21). (**B–C**) Representative 3D models, corresponding cross-sectional EM images, and 2D projections of dendritic skeletons are shown for the two morphological categories: unbranched (**B**) and branched (**C**). Blue bars mark the positions of cuticular bases. ab1D dendrites are pseudocolored in white in sample EM images. (**D**) Dendritic skeletons for the 13 segmented ab1D neurons with branched outer dendrites, with increasing branch numbers arranged from left to right.

for example). To investigate potential heterogeneity in ab1D dendritic morphologies, we focused on ab1 sensilla with high-quality images for further analysis.

We successfully generated 3D models for 21 ab1D neurons and found that their average soma volume was markedly smaller than that of ab1C ($23.73\ \mu m^3$ vs $42.68\ \mu m^3$). These models revealed significant heterogeneity in dendritic branching patterns, from unbranched to branched outer dendrites (*Figure 5*, inset). Among the 13 branched ab1D neurons, the primary branching point was located at various positions ranging from 8% to 44% above the sensillum base relative to the cuticular length (mean ± SD = 21 ± 10%, CV = 0.46). Dendritic termination, on the other hand, was less variable (CV = 0.09) and occurred between 63% and 97% of the cuticle length, with an average of 91 ± 8% (*Figure 5A*). Dendrites that began branching closer to the cuticle base did not necessarily terminate earlier (*Figure 5A*), differing from the relationship observed between ab1C flattening and dendritic termination locations (*Figure 3A*).

Interestingly, eight ab1D neurons had a simple, unbranched outer dendrite (*Figure 5B*), unlike most characterized basiconic ORNs, which typically have multiple dendritic branches (*Nava Gonzales et al., 2021*). In contrast, the remaining 13 ab1D neurons exhibited sparsely branched dendrites (*Figure 5C*). Notably, these neurons exhibited heterogeneity in both branching patterns and the number of branches, which ranged from two to seven (*Figure 5D*). This variability in dendritic branching patterns

mirrors the morphological heterogeneity observed in ab1C dendrites, suggesting that both types of neurons may have evolved diverse structural features to support specialized sensory functions.

## ab1D morphometrics

### Unbranched vs branched ab1D neurons

We then compared the morphometrics of unbranched outer dendrites with those of branched ones (*Figure 6A–E*, comparisons between two 'segment 1'). The summed measurements encompass the entire outer dendritic region where olfactory receptors are localized. The ab1D neurons with unbranched dendrites have an average outer dendritic length of 17.47 µm, sensory surface area of 13.29 µm$^2$, and volume of 0.69 µm$^3$. In comparison, ab1D with branched dendrites exhibited greater summed outer dendritic length (38.03 µm), surface area (19.74 µm$^2$), and volume (0.89 µm$^3$, *Table 1*). Notably, the dendritic surface area-to-volume ratio of neurons with branched dendrites was only about 14% higher than that of unbranched neurons (22.25 vs 19.50, *Table 1*). These comparisons suggest that within the ab1D neuronal population, branching markedly increases the total sensory surface area as much as 49% (19.74 µm$^2$ vs 13.29 µm$^2$), while providing only a modest enhancement in the overall surface area-to-volume ratio (*Table 1*).

### Branched ab1D neurons: proximal vs distal outer dendrites

For the branched ab1D neurons, we further compared the morphometrics of their proximal (unbranched) and distal (branched) outer dendritic segments (*Figure 6A–E*, comparisons between segments 2 and 3). Within a single neuron, the branched distal segment exhibited an average total length over 200% longer than the proximal unbranched segment, while showing a modest 20% increase in total surface area (*Figure 6A and B*). In addition, the distal segments had a significantly smaller total volume (distal/proximal ≈ 50%, *Figure 6C*), resulting in a much smaller average volume per unit length or cross-sectional area (distal/proximal ≈ 15%, *Figure 6D*). This indicates substantial narrowing of the branched outer dendrite, with the distal segment's average diameter being approximately 40% of the proximal segment's diameter. Consequently, the surface area-to-volume ratio of the distal branched segments was significantly higher—about 250% that of the proximal unbranched segments (*Figure 6E* and *Table 1*). In all, our analysis suggests that when comparing the distal and proximal dendritic segments for the same ab1D neuron, sparse branching (≤7 branches) has a modest impact on the overall sensory surface area but substantially increases the surface area-to-volume ratio in the outer dendritic region.

### Impacts of branch number on proximal dendritic morphometrics

Our morphometric analysis of 13 sparsely branched ab1D neurons allowed us to determine how branch number influences the morphometric properties of outer dendrites in an ORN population. First, we examined whether the distal dendritic branch number affects the morphometrics of the proximal unbranched segment and found no significant impact (*Figure 6F–J*). These findings indicate that when the branch number is low, the morphometrics of the proximal 'trunk' region remain relatively consistent.

### Impacts of branch number on distal dendritic morphometrics

In contrast to the proximal dendritic morphometrics, we found that the total length, surface area, and volume of distal segments scaled with the number of dendritic branches (*Figure 6K–M*). Interestingly, the diameter of individual dendritic branches and the surface area-to-volume ratio remained relatively consistent (*Figure 6N–O*), suggesting that the dimensions of individual dendritic branch are conserved regardless of the branch number. The average dimensions of an ab1D distal dendritic branch are 6.00 µm in length, 2.22 µm$^2$ in surface area, and 0.06 µm$^3$ in volume. Therefore, increasing the number of dendritic branches expands the total sensory surface area without altering the surface area-to-volume ratio, thereby preserving the dendrite's electrical signal propagation properties (see Discussion). Our analysis suggests that structural scaling in ab1D neurons may enhance sensory capacity while preserving the biophysical properties of dendrites.

## Auxiliary cells in the ab1 sensillum

In addition to ORNs, each olfactory sensillum contains three types of auxiliary cells: thecogen, trichogen, and tormogen. In previous work, we characterized the 3D morphology of these cells in

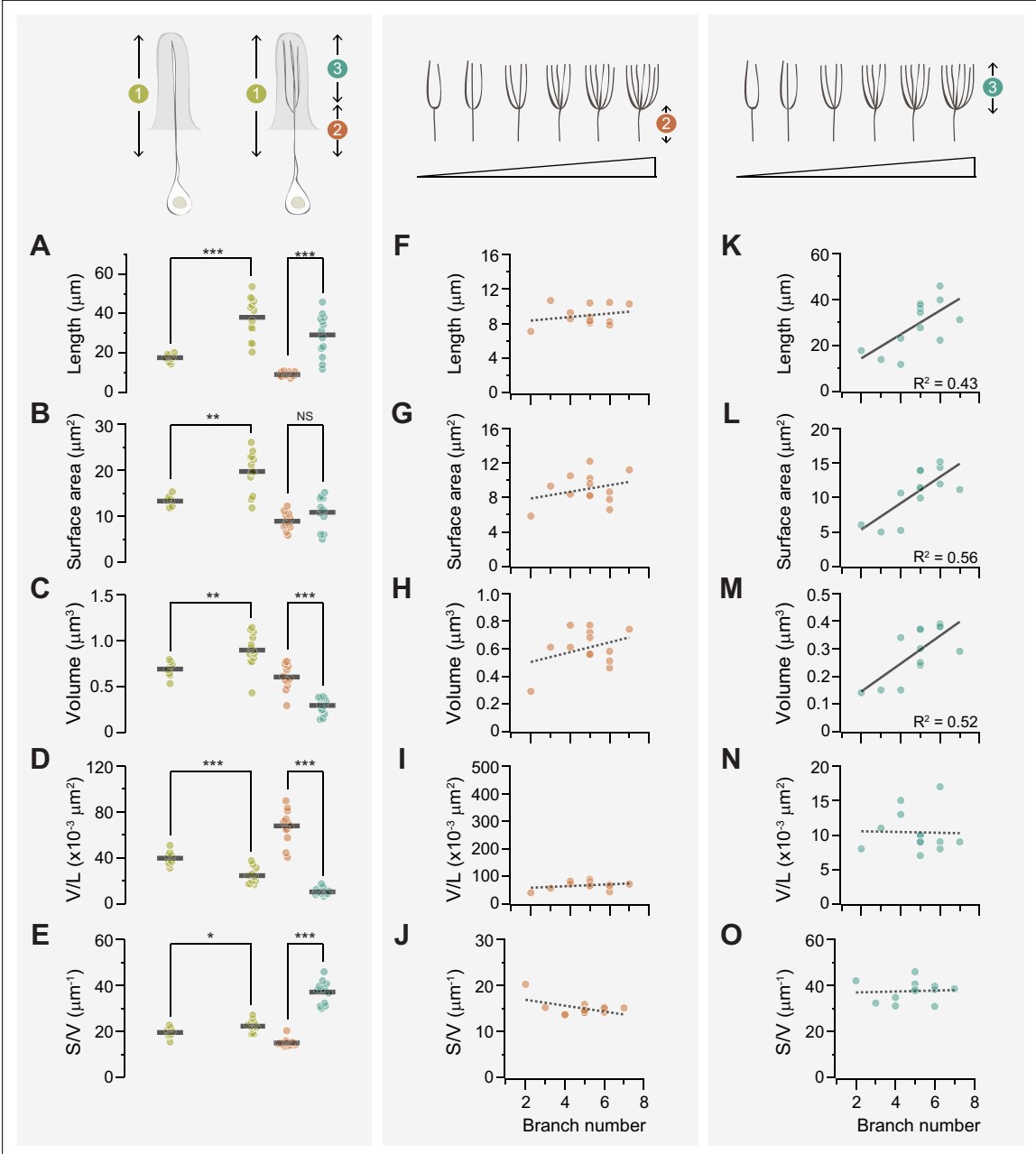

**Figure 6.** Branching enhances the surface area-to-volume ratio of the ab1D outer dendrite. (**A–E**) Quantitative comparisons of outer dendrites morphometric properties, including total length (**A**), surface area (**B**), volume (**C**), volume per unit length (**D**), and surface area-to-volume ratio (**E**). Data are presented for the entire outer dendrite (summed morphometrics, segment 1 in mustard), proximal unbranched (segment 2 in coral), and distal branched segments (segment 3 in teal). For branched ab1D neurons, segment 3 represents the summed morphometric measurements of all dendritic branches above the primary branch point, while segment 1 includes the combined values of segments 2 and 3. Each filled circle represents data from an individual neuron, with horizontal bars indicating the mean values (n=8 for unbranched neurons; n=13 for branched neurons). Statistical significance was determined using Mann-Whitney rank sum test for comparisons between two 'segment 1', and paired t-test for comparison between segments 2 and 3. *p<0.05; **p<0.01; ***p<0.005. (**F–J**) For branched ab1D dendrites, correlation analysis of the morphometric properties of the proximal outer dendrite (segment 2, coral) in relation to the number of dendritic branches. Parameters include the length (**F**), surface area (**G**), volume (**H**), volume per unit length (**I**), and surface area-to-volume ratio (**J**). (**K–O**) Similar to (**F–J**) but with the summed morphometric measurements of the distal outer dendritic segments (segment 3, teal). Linear fits are shown, with dashed lines indicating R²<0.4 and solid lines indicating R²>0.4. Also see *Figure 6—source data 1* for ab1D dendrite morphometrics.

The online version of this article includes the following source data for figure 6:

**Source data 1.** ab1D dendrite morphometrics.

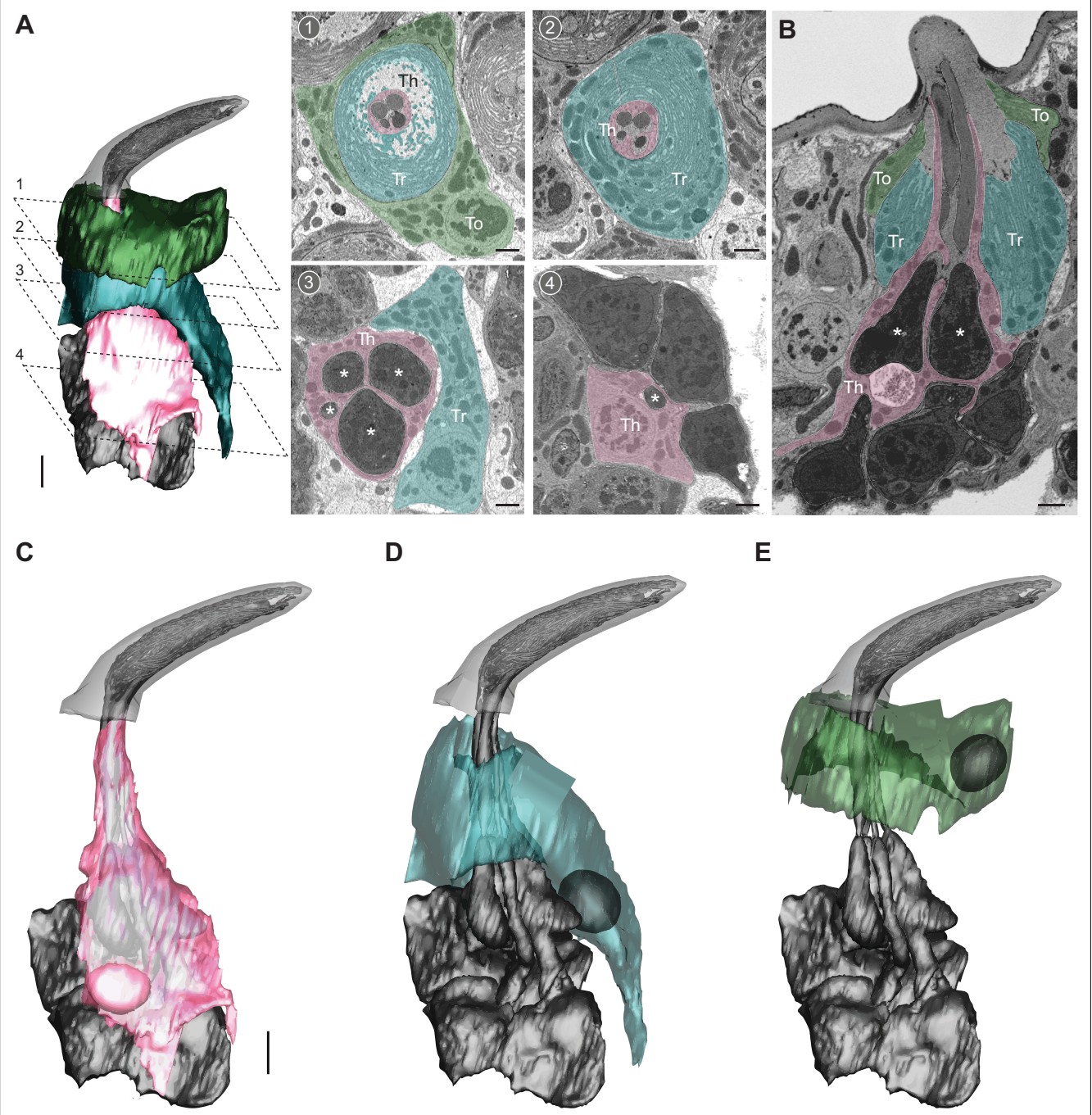

**Figure 7.** Auxiliary cells in the ab1 sensillum. (**A**) 3D model and serial block-face scanning electron microscopy (SBEM) images of an ab1 sensillum. Cells are pseudocolored to indicate identities: olfactory receptor neurons (ORNs) (gray), thecogen cell (pink, Th), trichogen cell (turquoise, Tr), tormogen cell (green, To). Dashed lines mark positions of the corresponding SBEM images: (1–2) outer dendritic region beneath the cuticle; (3) inner dendrites (asterisks); (4) ORN somas. (**B**) A longitudinal sensillum section rendered by IMOD. (C–E) 3D models of individual auxiliary cells: thecogen (**C**), trichogen (**D**), and tormogen (**E**). For simplicity, microlamellae of trichogen and tormogen cells were not segmented and thus not shown in 3D models. Scale bars: 2 µm for 3D models and 1 µm for SBEM images. See also *Video 3*.

the ab4 and ac3 sensilla (*Nava Gonzales et al., 2021*). To determine whether auxiliary cells exhibit sensillum-type-specific morphologies, we extended our analysis to the ab1 sensillum (*Figure 7A* and *Video 3*).

Similar to the ab4 and ac3 sensilla, the three auxiliary cells in the ab1 sensillum followed a conserved cellular organization. The thecogen cell exhibited an overall flat morphology, appearing to adhere

**Video 3.** 3D models of auxiliary cells in the ab1 sensillum.

https://elifesciences.org/articles/106389/figures#video3

closely to the ORNs. The cell formed a tight sleeve around the entire outer dendritic region beneath the cuticle base, as well as the inner dendrites and portions of the ORN somas. Notably, in the inner dendritic region, the thecogen cell separated individual ORNs from one another (*Figure 7A–C*). This differs from the ab4 sensillum, where the thecogen cell encloses both inner dendrites within a single bundle without separating them (*Nava Gonzales et al., 2021*).

The ab1 trichogen cell was positioned distal and lateral to the thecogen cell, with cellular processes surrounding the distal region of thecogen cell. The trichogen cell also featured an apical surface with extensive microlamellae bordering the sensillum-lymph cavity (*Figure 7A, B, and D*). The ab1 tormogen cell, the outermost of the three, partially enveloped the trichogen cell near the cuticle base. Its nucleus was located at the level of the ORN inner dendrites. Interestingly, unlike its counterparts in the ab4 and ac3 sensilla (*Nava Gonzales et al., 2021*), this tormogen cell lacked a stalk-like protrusion from its basal region, suggesting that this structure is not a universal feature of tormogen cells in insect olfactory sensilla.

## Discussion

Our nanoscale, population-wide analysis of the outer dendritic structures of ab1C and ab1D neurons revealed significant morphological diversity between different neuronal types and among homotypic ORNs. Between heterotypic ORNs, we identified a striking structural distinction between the $CO_2$-sensing ab1C neurons and the odor-sensing ab1D neurons. Specifically, ab1C neurons displayed flattened, sheet-like dendrites, in contrast to the cylindrical branches observed in ab1D neurons. Notably, most *Drosophila* ORNs, whether detecting food odors or fly pheromones, have outer dendrites with a cylindrical morphology (*Nava Gonzales et al., 2021*). Therefore, the sheet-like outer dendrites of ab1C neurons likely represent a unique structural adaptation for $CO_2$ or gas sensing in insects. Indeed, TEM studies in mosquitoes, moths, sand flies, and beetles have documented flattened dendritic lamellae, which are likely associated with $CO_2$-sensing neurons (*Hull and Cribb, 1997*; *Lu et al., 2007*; *McIver, 1972*; *McIver and Siemicki, 1975*; *Stange et al., 1995*; *Stange and Stowe, 1999*; *White et al., 1974*).

What might be the functional significance of ab1C's unique dendritic architectures? While both dendritic flattening and branching can expand sensory surface area and enhance SA/V ratio (*Figures 4 and 6*), the flattened ab1C dendritic sheets may facilitate better distribution or effective surface expression of the Gr21a and Gr63a receptors (*Jones et al., 2007*; *Kwon et al., 2007*; *Suh et al., 2004*). Notably, ectopic expression of Gr21a and Gr63a in the ab3A 'empty' neurons can confer $CO_2$ responsiveness, but the sensitivity is orders of magnitude lower than that of endogenous ab1C neurons (*Kwon et al., 2007*; *Yao and Carlson, 2010*), despite ab3A neurons having a much larger sensory surface area (ab3A: 145 $\mu m^2$; ab1C: 44 $\mu m^2$) (*Nava Gonzales et al., 2021* and this study). This observation suggests that the reduced $CO_2$ sensitivity in ab3A neurons may result from insufficient surface expression of Gr21a and Gr63a. Consistent with this, $CO_2$ sensitivity can be markedly enhanced by increasing the copy number of either Gr21a or Gr63a transgene, although not to the level observed in endogenous ab1C neurons (*Kwon et al., 2007*). In future research, it will be interesting to determine the molecular mechanisms underlying ab1C's flattened dendritic morphology and explore how selective manipulation of this feature affects $CO_2$ detection.

Beyond the morphological diversity observed between heterotypic ORNs, our study uncovered a rich spectrum of dendritic architectures within homotypic ab1C or ab1D neuron populations. For example, ab1C outer dendrites exhibit a wide range of morphologies, including plain sheets, tube-like structures enclosing several neighboring dendrites, split dendritic sheetlets, and combinations of all of these features (*Figure 3*). Similarly, ab1D dendrites range from simple, unbranched forms to numerously branched morphologies (*Figure 5*). These observations suggest that both neuron types may have evolved diverse structural features to support specialized sensory functions. One possibility

is that this diversity expands the range of sensory surface area among homotypic ORNs. In the ab1C population, surface area spans 1.78 folds, ranging from 30.67 $\mu m^2$ to 54.75 $\mu m^2$. In the ab1D population, the span is 2.21 folds, ranging from 11.77 $\mu m^2$ to 26.08 $\mu m^2$.

Future functional assays will be needed to determine whether the observed range of morphometric variation among homotypic *Drosophila* ORNs is sufficient to broaden their odor sensitivity, akin to the phenomenon observed in rodent homotypic ORNs, where varying odor sensitivities correlate with different cilia lengths (*Challis et al., 2015*; *Grosmaitre et al., 2006*). It would also be valuable to investigate whether the dendritic heterogeneity observed among homotypic adult ORNs represents dynamic morphological plasticity regulated in an activity-dependent manner, which may provide a structural mechanism for olfactory adaptation. These exciting open questions highlight the need for technological advances. In particular, linking nanoscale morphology to neuronal function would offer critical insights. However, this remains technically challenging, as it would require SBEM imaging of neurons previously recorded via single-sensillum electrophysiology. At present, no dye-labeling method is compatible with both single-sensillum recording and the cryofixation and sample preparation required for SBEM.

Alternatively, this morphological heterogeneity may have little functional significance and instead arise as a byproduct of the non-binary, graded nature of the molecular mechanisms underlying ORN-specific dendritic morphogenesis. For instance, in *C. elegans* amphid sensilla, the distinctive sensory cilia morphologies observed in AWA (tree-like branches), AWB (flattened, narrow protrusions), and AWC (flattened, wing-shaped) olfactory sensory neurons (*Doroquez et al., 2014*; *Perkins et al., 1986*) are the result of differential expression levels of a single immunoglobulin domain membrane protein, OIG-8 (*Howell and Hobert, 2017*; *Wallace and Shaham, 2017*). If a similar mechanism determines the dendritic morphologies of ab1C and ab1D neurons, the variability in morphogen expression levels—within the deterministic threshold specific to each neuronal type—could potentially explain the observed heterogeneity in dendritic morphology within the same ORN type.

Our morphometric analysis provides a wealth of data that paves the way for biologically realistic modeling in future studies. For example, one intriguing question involves the trade-offs of having extremely fine dendritic branches, such as the slender ab1D distal dendrite with a volume of 0.06 $\mu m^3$. Smaller volumes increase the SA/V ratio and amplify localized changes in ion concentrations, potentially enhancing signal conduction. For instance, in vertebrate olfactory cilia with an estimated volume of 0.5 $\mu m^3$, a 1 pA $Na^+$ influx can cause a significant ion flux of ~20 mM/s. However, a sharp $Na^+$ surge can inhibit $Ca^{2+}$ clearance via the $Na^+/Ca^{2+}/K^+$ exchanger, hampering response termination. Additionally, it causes a rapid rise in osmotic pressure, increasing the risk of ciliary swelling and damage. This constraint likely explains why vertebrate ORNs rely on $Cl^-$ efflux rather than $Na^+$ influx as the primary transduction current (*Reisert and Reingruber, 2019*). In contrast, *Drosophila* ORNs lack reported $Cl^-$ efflux currents and instead rely on cation influx for olfactory transduction (*Benton, 2022*). Modeling the biophysical effects of this influx on fine outer dendrites would be valuable. Future studies should also compare the biophysical implications of dendritic flattening vs branching and explore the functional impact of the 'dendrite-within-dendrite' phenomenon in ab1C. How does this unique structure affect odor detection and ephaptic coupling among neighboring neurons? These questions highlight exciting avenues for further research.

## Materials and methods
### Animals

*D. melanogaster* flies were reared on standard cornmeal food containing molasses at 25°C, ~60% humidity under a 12 hr light/dark cycle in an incubator. To facilitate comparison with our published ORN morphometrics data (*Nava Gonzales et al., 2021*), flies of the same genotype (*Or7a-GAL4>10xUAS-myc-APEX2-Orco*) were used to generate the antennal volume. The fly stock numbers are BDSC 91810 for *Or7a-GAL4* (*Lin et al., 2015*), *and BDSC 79214 for 10xUAS-myc-APEX2-Orco* (*Tsang et al., 2018*). However, APEX2-mediated DAB staining was not applied in this study to label specific neurons. Experimental flies were collected upon eclosion, separated by sex, and co-housed in groups of 10. Female flies aged 6–8 days were used for experiments.

Antenna dissection was performed as follows. A fly was first wedged into the narrow end of a truncated plastic 200 µl pipette tip to expose the antenna, which was subsequently stabilized between

a tapered glass microcapillary and a coverslip covered with double-sided tape. To facilitate solution exchange during sample preparation, a sharp glass electrode was used to puncture the lateral side of the antenna, a region with a low density of the ab1 sensillum type. Finally, the third antennal segment was severed from the fly's head by pinching the second segment with fine forceps.

## Tissue preparation and SBEM volume acquisition

The SBEM volume of the *D. melanogaster* antenna was generated following the CryoChem protocol (*Tsang et al., 2018*). Briefly, dissected antennae were immediately subject to high-pressure freezing in a solution of 0.15 M sodium cacodylate and 20% BSA using a high-pressure freezing machine (Bal-Tec HPM 010). The frozen samples were then transferred to a freeze-substitution solution containing 0.2% glutaraldehyde (#18426, Ted Pella, CA, USA), 0.1% uranyl acetate (Electron Microscopy Sciences, USA), and 1% water in acetone (#AC326800010, ACROS Organics, USA) in a liquid nitrogen bath. Freeze substitution was performed in a Leica EM AFS2 device held at –90°C for 58 hr, from –90°C to –60°C for 15 hr, at –60°C for 15 hr, from –60°C to –30°C for 15 hr, and then at –30°C for 15 hr. During the final hour at –30°C, samples were washed three times (20 min each) in an acetone solution containing 0.2% glutaraldehyde and 1% water, before being transferred to ice for 1 hr prior to rehydration (see below).

The cryofixed samples were gradually rehydrated in a series of nine rehydration solutions, each for 10 min on ice.

1. 5% water, 0.2% glutaraldehyde in acetone
2. 10% water, 0.2% glutaraldehyde in acetone
3. 20% water, 0.2% glutaraldehyde in acetone
4. 30% water, 0.2% glutaraldehyde in acetone
5. 50% 0.1 M HEPES (Gibco, Taiwan), 0.2% glutaraldehyde in acetone
6. 70%, 0.1 M HEPES, 0.2% glutaraldehyde in acetone
7. 0.1 M HEPES
8. 0.1 M sodium cacodylate with 100 mM glycine
9. 0.1 M sodium cacodylate

After rehydration, the samples were subject to en bloc heavy metal staining in a solution of 2% $OsO_4$, 1.5% potassium ferrocyanide, and 2 mM $CaCl_2$ in 0.1 M sodium cacodylate for 1 hr at room temperature. Then, samples were washed five times with water (5 min per wash) before being incubated in 0.5% thiocarbohydrazide (Electron Microscopy Sciences, USA) for 30 min at room temperature. Following another series of water washes, the samples were incubated in 2% $OsO_4$ for 30 min at room temperature. After a final rinse with water, the samples were transferred to 2% aqueous uranyl acetate (filtered with 0.22 µm filter) at 4°C overnight. The samples were then washed with water and then subjected to the dehydration steps described below.

Dehydration was performed in six consecutive 10 min steps: 70% ethanol, 90% ethanol, 100% ethanol (twice), followed by 100% acetone (twice). All ethanol steps were conducted on ice. The first acetone step used ice-cold acetone, while the second was performed with acetone at room temperature.

Resin infiltration was carried out in a 1:1 solution of Durcupan ACM resin and acetone and incubated overnight on a shaker. The samples were then transferred in fresh 100% Durcupan ACM resin twice, with a 6–7 hr interval between transfers. During incubation in 100% resin, the samples were placed in a vacuum chamber on a rocker to facilitate the evaporation of residual acetone. After an overnight incubation in 100% resin, the samples were embedded in fresh resin and polymerized at 60°C for at least 2 days. The composition of Durcupan ACM resin (Sigma-Aldrich) was 11.4 g component A, 10 g component B, 0.3 g component C, and 0.1 g component D.

Microcomputed X-ray tomography was used to determine the position and proper orientation of the resin-embedded specimens. Samples were mounted on aluminum pins with conductive silver epoxy (Ted Pella) and sputter coated with gold-palladium for SBEM imaging with a Gemini SEM 300 (Zeiss) equipped with a Gatan 3View 2XP microtome system and the OnPoint backscatter detector.

The antennal SBEM volume was acquired at 2.5 kV using a 30 µm aperture, with the electron gun set to analytic mode and the beam operating in high-current mode. Nitrogen gas was used for focal charge compensation to reduce charging artifacts. Imaging was performed with a dwell time of 1 µs, a pixel size of 5 nm, and a Z-step of 40 nm. The X and Y pixel numbers were 1909 and 2061,

respectively, and there were a total of 2571 Z slices. After data collection, the images were converted to MRC format, and rigid alignment of the image slices was performed using cross-correlation in the IMOD image processing package (https://bio3d.colorado.edu/imod/). The SBEM volume is available in the Cell Image Library (https://www.cellimagelibrary.org/) with the accession numbers CIL:57519.

## Image segmentation

In the fly antenna volume, the ab1 sensilla were identified based on their characteristic size and the presence of four neurons, as ab1 is the only large basiconic sensillum type housing four ORNs (*Nava Gonzales et al., 2021*; *Shanbhag et al., 1999*). Manual segmentation was conducted using IMOD's drawing tools by placing closed contours around the structures of interest in serial sections (*Kremer et al., 1996*). The sensillum cuticle, ORN soma, and inner and outer dendritic segments were saved as distinct objects to facilitate morphometric measurement of individual structures. The ciliary constriction was used to define the boundary between the inner and outer dendrites (*Shanbhag et al., 2000*).

For the subset of ab1C neurons featuring fully curled, tube-like outer dendrites, the enclosed dendritic regions were segmented into two separate objects: the hollow inner region and the membrane-bound outer region. Subsequently, all segmented objects were 'meshed' to connect adjacent contours, creating continuous 3D structures. Detailed information about 'imodmesh' and IMOD's drawing tools is available in the IMOD user guide (https://bio3d.colorado.edu/imod/doc/man/imod-mesh.html; https://bio3d.colorado.edu/imod/doc/3dmodHelp/plughelp/drawingtools.html). All 3D neuron models generated in this study are available in the NIH 3D repository (https://3d.nih.gov/) under the entry ID: 3DPX-021684.

## SBEM image post-processing

For representative SBEM images, image quality was enhanced using the DenoiseEM plug-in for ImageJ, which offers multiple denoising algorithm options. Briefly, TIFF images were first loaded into ImageJ and converted to a 16-bit file format. Multiple regions of interest within the sensillar lumen were sampled to train the denoising algorithms, and the optimal algorithm was selected based on the best signal-to-blur ratio or overall image quality. For the SBEM images presented in this study, the Gaussian algorithm was most frequently used. To further enhance the visibility of dendritic branches, the contrast and brightness of the denoised images were adjusted in ImageJ. The final images were then converted back to RGB format and exported as TIFF files. Detailed information about DenoiseEM is available in the DenoiseEM plug-in page (https://bioimagingcore.be/DenoisEM/).

## 3D model videos

Movies for each 3D ORN model were created using IMOD's 'Movie Montage Control' and 'Movie Sequence Dialog' functions. Briefly, 'Movie Montage Control' allows users to manipulate 3D models and record these manipulations or scenes as a series of images using the 'Set Start' and 'Set End' controls. IMOD then interpolates a user-specified number of frames between the two views to create a smooth transition. This workflow supports creating multiple scenes (e.g. zooming in/out, rotations, or displaying image planes) as a sequence using the 'Movie Sequence Dialog'. Each scene in the sequence was exported as TIFF images. These images were then compiled into an image sequence in QuickTime and saved as a .mov file at 60 frames per second. Detailed information about IMOD's movie and montage controls is available in the IMOD user guide (https://bio3d.colorado.edu/imod/doc/3dmodHelp/modelMovie.html).

## Skeletonization

To visualize dendritic branching patterns, the 3D models of ORN dendrites in MOD format were first converted to VRML2 files using the command 'imod2vrml2' in IMOD. The VRML2 files were then imported into Amira (2020.2 version; Thermo Fisher Scientific, USA) and converted into a binary mesh, with the 3D model area colored in white and the background in black. The AutoSkeleton module in Amira was used to identify the center of mass within each mesh region and generate a 2D skeleton. The skeletons were then manually edited using the 'Filament editor' in Amira by overlaying them with ORN 3D models to correct errors such as extra loops or branches.

These skeletons, in SWC format, were imported into neuTube software (https://www.neutracing.com/), where dendritic branches were manually spread onto a 2D plane. Briefly, a primary branch and

all its downstream branches were first selected to allow all the branches to be edited and moved as a group. This process was repeated for secondary, tertiary, and higher-order branches until overlap between branches was minimized.

## Morphometric analysis

For morphometric analysis, the sensillum cuticle, ORN soma, inner dendrite, and the proximal and distal outer dendritic segments were analyzed as separate objects.

### Surface area and volume

The morphometric values were extracted from individual objects using the 'imodinfo' function in IMOD. Detailed information about 'imodinfo' is available in the IMOD user guide (https://bio3d.colorado.edu/imod/doc/man/imodinfo.html). Among the three volume measurement options in 'imodinfo', the 'Volume Inside Mesh' option was selected to measure ORN volumes. For enclosed tube-like structures in the ab1C outer dendrites, volume was calculated by subtracting the inner object volume from the outer object volume, while surface area was determined by summing the surface areas of the inner and outer objects. For the branched outer dendrites of ab1D, the total volume and surface area were obtained by summing the values from individual branches.

### Absolute length

To calculate the absolute lengths of individual objects, these structures were first skeletonized using Amira. The resulting SWC files were imported into R, where pixel coordinates were scaled to micrometers using scaling factors derived from the 'imodinfo' command. The length of each component was then calculated using the Pythagorean theorem.

### Relative position to sensillum cuticle

To calculate the relative positions of dendritic branching, flattening, or terminal points with respect to the cuticle, the previously mentioned SWC skeletons were first imported into Python. Branching points and dendritic termini were extracted using the PyNeuroML package (https://pyneuroml.readthedocs.io/en/development/). Each point of interest (e.g. ciliary constriction, branching or flattening point, and dendritic terminus) was projected onto the nearest point on the cuticle skeleton. The cuticle proportion of a point was defined as its relative position along the cuticle, scaled from 0 (cuticle base) to 1 (cuticle tip). For points below the cuticle base, the bottom segment of the cuticle skeleton was extrapolated, and the point was projected onto this segment. In such cases, the cuticle proportion was scaled from 0 (at the cuticle base) to negative infinity, with –1 representing one cuticle length down the extrapolated segment.

### Total number of dendrites in ab1 sensilla

The dendritic branch count for each ab1 sensillum, as shown in *Figure 1B*, was estimated using an image slice taken at approximately the midpoint of the sensillum cuticle. This image plane was rotated to a perpendicular orientation relative to the long axis of the cuticle using IMOD's slicer tool. In this orientation, the images were exported in TIFF format, denoised, and branches were manually counted. The sensillum cross-sectional area at the midpoint was measured by outlining the sensillar lumen with a contour and calculating its area using the 'imodinfo' command in IMOD. For more information about IMOD's slicer tool, refer to the IMOD user guide (https://bio3d.colorado.edu/imod/doc/3dmodHelp/slicer.html).

## Statistics

All values were presented as mean ± standard deviation (SD). Paired two-tailed t-tests were used for morphometric comparisons between neighboring ORNs within the same sensillum. For comparisons between non-neighboring neurons, rank sum tests or unpaired two-tailed t-tests (if the Shapiro-Wilk normality test was passed) were applied. A p-value of <0.05 was considered statistically significant.

## Acknowledgements

We thank Nabeeha Rashid, Tulio Magana, Pawel Vijayakumar, and Julissa Meza for assistance with SBEM image segmentation, and Renny Ng for comments on the manuscript. This study was supported by NIH grants R01DC016466, R01DC021551, R21AI169343, and R21DC020536 (C-YS); U24NS120055, and R01GM138780 (MHE).

## Additional information

### Funding

| Funder | Grant reference number | Author |
| --- | --- | --- |
| National Institute of Allergy and Infectious Diseases | R21AI169343 | Chih-Ying Su |
| National Institute on Deafness and Other Communication Disorders | R01DC016466 | Chih-Ying Su |
| National Institute on Deafness and Other Communication Disorders | R01DC021551 | Chih-Ying Su |
| National Institute on Deafness and Other Communication Disorders | R21DC020536 | Chih-Ying Su |
| National Institute of Neurological Disorders and Stroke | U24NS120055 | Mark H Ellisman |
| National Institute of General Medical Sciences | R01GM138780 | Mark H Ellisman |

The funders had no role in study design, data collection and interpretation, or the decision to submit the work for publication.

### Author contributions

Jonathan Choy, Shadi Charara, Data curation, Formal analysis, Validation, Investigation, Visualization, Methodology, Writing – original draft, Writing – review and editing; Kalyani Cauwenberghs, Software, Formal analysis, Investigation, Methodology, Writing – original draft, Writing – review and editing; Quintyn McKaughan, Investigation, Visualization, Methodology; Keun-Young Kim, Methodology; Mark H Ellisman, Resources, Funding acquisition, Methodology; Chih-Ying Su, Conceptualization, Data curation, Formal analysis, Supervision, Funding acquisition, Validation, Investigation, Visualization, Writing – original draft, Project administration, Writing – review and editing

### Author ORCIDs

Jonathan Choy ⬤ https://orcid.org/0009-0000-3962-5081
Chih-Ying Su ⬤ https://orcid.org/0000-0002-0005-1890

Reviewer #1 (Public review): https://doi.org/10.7554/eLife.106389.3.sa1
Reviewer #2 (Public review): https://doi.org/10.7554/eLife.106389.3.sa2
Reviewer #3 (Public review): https://doi.org/10.7554/eLife.106389.3.sa3
Author response https://doi.org/10.7554/eLife.106389.3.sa4

## Additional files

### Supplementary files

MDAR checklist

## Data availability

All data generated or analyzed during this study are included in the manuscript and supporting files; source data files have been provided for Figures 1, 2, 4, and 6. All 3D neuron models generated in this study are available in the NIH 3D repository (https://3d.nih.gov/) under the entry ID: 3DPX-021684.

The following dataset was generated:

| Author(s) | Year | Dataset title | Dataset URL | Database and Identifier |
|---|---|---|---|---|
| Choy J, Charara S, Su C-Y | 2025 | Co-housed CO2-sensing and odor-sensing neurons in *Drosophila melanogaster* | https://3d.nih.gov/entries/3DPX-021684 | NIH 3D, 3DPX-021684 |

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
