## [Editor Report · eLife Assessment]

This **valuable** study reveals surprising morphological diversity of *Drosophila* sensory neurons. Using serial block-face electron microscopy, the authors created detailed 3D reconstructions of large neuronal populations, **convincingly** finding significant structural variation both within and across distinct classes. These results form the basis for testable hypotheses on how neuronal arborization is optimized for particular sensory functions. This research will be highly relevant to biologists in the fields of physiology, insect chemosensation, and neuroscience.

---

## [Referee Report · Reviewer #1 (Public review)]

The authors of this study use electron microscopy and 3D reconstruction techniques to study the morphology of distinct classes of *Drosophila* sensory neurons *across many neurons of the same class.* This is a comprehensive study attempting to look at nearly all the sensory neurons across multiple sensilla in the same animal to determine (a) how much morphological variability exists between and within neurons of different and similar sensory classes and (b) identify dendritic features that may have evolved to support particular sensory functions. This study builds upon the authors' previous work which allowed them to identify and distinguish sensory neuron subtypes in the EM volumes without additional staining so that reconstructed neurons could reliably be placed in the appropriate class. This work is unique in looking at a large number of individual neurons of the same class to determine what is consistent and what is variable about their class-specific morphologies.

This means that in addition to providing specific structural information about these particular cells, the authors explore broader questions of how much morphological diversity exists between sensory neurons of the same class. This then informs our conceptualization about how different dendritic morphologies might affect specific sensory and physiological properties of neurons.

The authors found that CO2 sensing neurons have an unusual, sheet-like morphology in contrast to the thin branches of odor-sensing neurons. They show that this morphology greatly increases the surface area to volume ratio above what could be achieved by modest branching of thin dendrites, and posit that this might be important for their sensory function, though this was not directly tested in their study due to technical limitations. The study is mainly descriptive in nature, but thorough, and provides a nice jumping off point for future functional studies. One interesting future analysis could be to examine all four cell types within a single sensilla together to see if there are any general correlations that could reveal insights about how morphology is determined and relative contributions of intrinsic mechanisms vs interactions with neighboring cells. For example, if higher-than-average branching in one cell type correlated with higher-than-average branching in another type when within the same sensilla, it might suggest differential amounts of extracellular growth or branching cues within a given sensillum drive any heterogeneity observed within a class across sensilla. Conversely, if higher branching in one cell type consistently leads to reduced length or branching of the other neurons within its sensillum, this might point to dendrite-dendrite interactions between cells undergoing competitive or repulsive interactions to define territories within each sensillum as a major determinant of the variability.

Strengths:

This work provides a thorough morphometric analysis of the neurons of the *majority of all ab1 sensilla* across a single antenna. The authors use this analysis to (1) characterize the unique dendritic architecture of ab1C neurons relative to other ORNs including ab1D and (2) provide evidence of substantial morphological diversity even within a single subclass of neuron.

Weaknesses:

This is primarily a descriptive paper due to technical limitations since it is not currently technically feasible to determine individual ORN response properties and tie them to identified neurons with detailed EM-based ultrastructural analyses, nor to predictably alter dendritic morphology of these cells to directly test how different morphologies affect sensory function. However, the quantitative descriptive findings presented here will shape these future questions and are necessary for any such future work.

---

## [Referee Report · Reviewer #2 (Public review)]

Summary:

The manuscript employs serial block‐face electron microscopy (SBEM) and cryofixation to obtain high‐resolution, three‐dimensional reconstructions of *Drosophila* antennal sensilla containing olfactory receptor neurons (ORNs) that detect CO2. This method has been used previously by the same lab in Gonzales et. al, 2021. (https://elifesciences.org/articles/69896), and Zhang et. al, 2019 Nature Communications. The previous study by Zhang also correlated morphometric measurements from SBEM with asymmetric ephaptic activity for paired neurons using electrophysiology across multiple olfactory sensilla. This manuscript applies the same SBEM method to now characterize the ab1 sensillum which houses the ab1C, CO2 detecting neuron, but stops short of integration neuronal activity with structural variability.

The SBEM-based morphometric studies do however significantly advance preliminary observations from older two-dimensional TEM-based reports. Previous images of the putative CO2 neuron in *Drosophila* (Shanbhag et al., 1999) and in mosquitoes (McIver and Siemicki, 1975; Lu et al, 2007) reported that the dendritic architecture of the CO2 neuron was somewhat different (circular and flattened, lamellated) from other olfactory neurons in the antenna of insects. In this study, the authors confirm this different morphology but also classify it into distinct subtypes (loosely curled, fully curled, split, and mixed).

Strengths:

The study makes a convincing case that ab1C neurons exhibit a unique, dendritic morphology unlike the canonical cylindrical dendrites found in ab1D neurons. This observation extends previous qualitative TEM findings by not only confirming the presence of flattened lamellae in CO₂ neurons but also quantifying key morphometrics such as dendritic length, surface area, and volume, and calculating surface area-to-volume ratios. The enhanced ratios observed in the flattened segments are speculated to be linked to potential advantages in receptor distribution (e.g., Gr21a/Gr63a) and efficient signal propagation.

Weaknesses:

Although this quantitative approach is very robust compared to earlier reports, interpretations are somewhat limited by the absence of direct electrophysiological data to confirm whether ultrastructural differences translate into altered neuronal function. The biggest question remains unanswered: whether structural variation observed in the ab1C dendrites by SBEM have an electrophysiological functional relevance?

Surveys of ab1 sensillum with single-sensillum recordings (even a few from multiple *Drosophila* antenna) as they have done for ab2s and others in the past, would have measured spontaneous activity, spike amplitude, and response to CO2. This could have allowed for comparison of frequency of functional variation, if any, to structural variation and a discussion would therefore have strengthened the overall characterization. In the case of ab2 sensilla the authors find very little variance, could the ab1 also be the same? In the absence of this data, it becomes hard to speculate whether structural variation observed in the ab1C dendrites by SBEM have any functional relevance or whether they are simply random variations in dendrite development.

Additionally, artifacts could be a consideration, even though Cryofixation is superior to chemical fixation. Although this is hard to address, all types of fixations in TEMs cause some artifacts, as does serial sectioning. An understanding of the error rates for the SBEM method would have increased the confidence in the conclusions drawn. For example, what is the structural variation of SBEMs in the ab2 population, which shows very little electrophysiological variation? Can a comparison be done?

---

## [Referee Report · Reviewer #3 (Public review)]

Summary:

In the current manuscript entitled "Population-level morphological analysis of paired CO2- and odor-sensing olfactory neurons in *D. melanogaster* via volume electron microscopy", Choy, Charara et al. use volume electron microscopy and neuron reconstruction to compare the dendritic morphology of ab1C and ab1D neurons of the *Drosophila* basiconic ab1 sensillum. They aim to investigate the degree of dendritic heterogenity within a functional class of neurons using ab1C and ab1D, which they can identify due to the unique feature of ab1 sensilla to house four neurons and the stereotypic location on the third antennal segment. This is a great use of volumetric electron imaging and neuron reconstruction to sample a population of neurons of the same type. Their data convincingly shows that there is dendritic heterogenity in both investigated populations and their sample size is sufficient to strongly support this observation. This data proposes that the phenomenon of dendritic heterogenity is common in the *Drosophila* olfactory system and will stimulate future investigations into the developmental origin, functional implications and potential adaptive advantage of this feature.

Moreover, the authors discovered that there is a difference between CO2- and odour sensing neurons of which the first show a characteristic flattened and sheet-like structure not observed in other sensory neurons sampled in this and previous studies. They hypothesize that this unique dendritic organization which increases the surface area to volume ratio, might allow more efficient Co2 sensing by housing higher numbers of Co2 receptors. This is supported by previous attempts to express Co2 sensors in olfactory sensory neurons which lack this dendritic morphology, resulting in lower Co2 sensitivity compared to endogenous neurons.

Overall, this detailed morphological description of olfactory sensory neurons' dendrites convincingly shows heterogeneity in two neuron classes with potential functional impacts for odour sensing.

Strength:

The volumetric EM imaging and reconstruction approach offers unpreceeded details in single cell morphology and compares dendrite heterogenity across a great fraction of ab1 sensilla.

The authors identify specific shapes for ab1C sensilla potentially linked to their unique function in CO2 sensing.

Weaknesses:

While the morphological description is highly detailed, current methods prevent linking morphology to odour sensitivity or other properties of the neurons. Therefore, this study remains mainly descriptive and will require future work to link neuron structure and function.

---

## [Author Response]

The following is the authors’ response to the original reviews

**Reviewer #1 (Public Review):**
The authors of this study use electron microscopy and 3D reconstruction techniques to study the morphology of distinct classes of *Drosophila* sensory neurons *across many neurons of the same class.* This is a comprehensive study attempting to look at nearly all the sensory neurons across multiple sensilla to determine (a) how much morphological variability exists between and within neurons of different and similar sensory classes, and (2) identify dendritic features that may have evolved to support particular sensory functions. This study builds upon the authors' previous work, which allowed them to identify and distinguish sensory neuron subtypes in the EM volumes without additional staining so that reconstructed neurons could reliably be placed in the appropriate class. This work is unique in looking at a large number of individual neurons of the same class to determine what is consistent and what is variable about their class-specific morphologies.

This means that in addition to providing specific structural information about these particular cells, the authors explore broader questions of how much morphological diversity exists between sensory neurons of the same class and how different dendritic morphologies might affect sensory and physiological properties of neurons.

The authors found that CO2-sensing neurons have an unusual, sheet-like morphology in contrast to the thin branches of odor-sensing neurons. They show that this morphology greatly increases the surface area to volume ratio above what could be achieved by modest branching of thin dendrites, and posit that this might be important for their sensory function, though this was not directly tested in their study. The study is mainly descriptive in nature, but thorough, and provides a nice jumping-off point for future functional studies. One interesting future analysis could be to examine all four cell types within a single sensilla together to see if there are any general correlations that could reveal insights about how morphology is determined and the relative contributions of intrinsic mechanisms vs interactions with neighboring cells. For example, if higher than average branching in one cell type correlated with higher than average branching in another type, if in the same sensilla. This might suggest higher extracellular growth or branching cues within a sensilla. Conversely, if higher branching in one cell type consistently leads to reduced length or branching in another, this might point to dendrite-dendrite interactions between cells undergoing competitive or repulsive interactions to define territories within each sensilla as a major determinant of the variability.

We thank the reviewer for the insightful comments and appreciation for our study.

**Reviewer #2 (Public Review):**
The manuscript employs serial block‐face electron microscopy (SBEM) and cryofixation to obtain high‐resolution, three‐dimensional reconstructions of *Drosophila* antennal sensilla containing olfactory receptor neurons (ORNs) that detectCO2. This method has been used previously by the same lab in Gonzales et. al, 2021. (https://elifesciences.org/articles/69896), which had provided an exemplary model by integrating high-resolution EM with electrophysiology and cell-type-specific labeling.

We thank the reviewer for expressing appreciation for our published study.

The previous study ended up correlating morphology with activity for multiple olfactory sensillar types. Compared to the 2021 study, this current manuscript appears somewhat incomplete and lacks integration with activity.

We thank the reviewer for their feedback. However, we would like to clarify that our previous study did not correlate morphology with activity to a greater extent than the current study. Both employed the same cryofixation, SBEM-based approach without recording odor-induced activity, but the focus of the current work is fundamentally different. While the previous study examined multiple sensillum types, the current study concentrates on a single sensillum type to address a distinct biological question regarding morphological heterogeneity. We appreciate the opportunity to clarify this distinction, and we hope that the revised manuscript more clearly conveys the unique scope and contributions of this study.

In fact older studies have also reported two-dimensional TEM images of the putative CO2 neuron in *Drosophila* (Shanbhag et al., 1999) and in mosquitoes (McIver and Siemicki, 1975; Lu et al, 2007), and in these instances reported that the dendritic architecture of the CO2 neuron was somewhat different (circular and flattened, lamellated) from other olfactory neurons.

We thank the reviewer for pointing this out. As noted in both the Introduction and Discussion sections, previous studies—including those cited by the reviewer—suggested that CO2-sensing neurons may have a distinct dendritic morphology. However, those earlier studies lacked the means to definitively link the observed morphology to CO2 neuron identity.

In contrast, our study assigns neuronal identity based on quantitative morphometric measurements, allowing us to confidently associate the unique dendritic architecture with CO2 neurons. Furthermore, we extend previous observations by providing full 3D reconstructions and nanoscale morphometric analyses, offering a much more comprehensive and definitive characterization of these neurons. We believe this represents a significant advancement over earlier work.

The authors claim that this approach offers an artifact‐minimized ultrastructural dataset compared to earlier. In this study, not only do they confirm this different morphology but also classify it into distinct subtypes (loosely curled, fully curled, split, and mixed). This detailed morphological categorization was not provided in prior studies (e.g., Shanbhag et al., 1999).

We thank the reviewer for acknowledging the significance of our study.

The authors would benefit from providing quantitative thresholds or objective metrics to improve reproducibility and to clarify whether these structural distinctions correlate with distinct functional roles.

We thank the reviewer for raising this point. However, we would like to clarify that assigning neurons to strict morphological subtypes was not the primary aim of our study. In practice, dendritic architectures can be highly complex, with individual neurons often displaying features characteristic of multiple subtypes. This is precisely why we included a “mixed” subtype category—to acknowledge and capture this morphological heterogeneity rather than impose rigid classification boundaries.

Our intent in defining subtypes was not to imply discrete functional classes, but rather to highlight the range of morphological variation observed across ab1C neurons. While we agree that exploring potential correlations between structure and function is an important future direction, the current study focuses on characterizing this diversity using 3D reconstruction and morphometric analysis. We hope this clarifies the purpose and scope of our morphological categorization.

Strengths:The study makes a convincing case that ab1C neurons exhibit a unique, flattened dendritic morphology unlike the cylindrical dendrites found in ab1D neurons. This observation extends previous qualitative TEM findings by not only confirming the presence of flattened lamellae in CO₂ neurons but also quantifying key morphometrics such as dendritic length, surface area, and volume, and calculating surface area-to-volume ratios. The enhanced ratios observed in the flattened segments are speculated to be linked to potential advantages in receptor distribution (e.g., Gr21a/Gr63a) and efficient signal propagation.

We thank the reviewer for appreciating the significance our current study.

Weaknesses:While the manuscript offers valuable ultrastructural insights and reveals previously unappreciated heterogeneity among CO₂-sensing neurons, several issues warrant further investigation in addition to the points made above.(1) Although this quantitative approach is robust compared to earlier descriptive reports, its impact is somewhat limited by the absence of direct electrophysiological data to confirm that ultrastructural differences translate into altered neuronal function. A direct comparison or discussion of how the present findings align with the functional data obtained from electrophysiology would strengthen the overall argument.

We thank the reviewer for this comment. We would like to clarify, however, that our study does not claim that the observed morphological heterogeneity necessarily leads to functional diversity. Rather, we consider this as a possible implication and discuss it as a potential question for future research. This idea is raised only in the Discussion section, and we are carefully not to present functional diversity as a conclusion of our study. Nonetheless, we have reviewed the relevant paragraph to ensure the language remains cautious and does not overstate our interpretation.

We also acknowledge the significance of directly linking ultrastructural features to neuronal function through electrophysiological recordings. However, at present, it is technically challenging to correlate the nanoscale morphology of individual ORNs with their functional activity, as this would require volume EM imaging of the very same neurons that were recorded via electrophysiology. Currently, there is no dye-labeling method compatible with single-sensillum recording and SBEM sample preparation that allows for unambiguous identification and segmentation of recorded ORNs at the necessary ultrastructural resolution.

To acknowledge this important limitation, we have added a paragraph in the Discussion section, as suggested, to clarify the current technical barriers and to highlight this as a promising direction for future methodological advances.

(2) Clarifying the criteria for dendritic subtype classification with quantitative parameters would enhance reproducibility and interpretability. Moreover, incorporating electrophysiological recordings from ab1C neurons would provide compelling evidence linking structure and function, and mapping key receptor proteins through immunolabeling could directly correlate receptor distribution with the observed morphological diversity.

Please see our response to the comment regarding the technical limitations of directly correlating ultrastructure with electrophysiological data.

In addition, we would like to address the suggestion of using immunolabeling to map receptor distribution in relation to the 3D EM models. Currently, antibodies against Gr21a or Gr63a (the receptors expressed in ab1C neurons) are not available. Even if such antibodies were available, immunogold labeling for electron microscopy requires harsh detergent treatment to increase antibody permeability, damaging morphological integrity. These treatments would compromise the very morphological detail that our study aims to capture and quantify.

(3) Even though Cryofixation is claimed to be superior to chemical fixation for generating fewer artifacts, authors need to confirm independently the variation observed in the CO2 neuron morphologies across populations. All types of fixation in TEMs cause some artifacts, as does serial sectioning. Without understanding the error rates or without independent validation with another method, it is hard to have confidence in the conclusions drawn by the authors of the paper.

We thank the reviewer for raising concerns regarding potential artifacts in morphological analyses. However, we would like to clarify that cryofixation is widely regarded as a gold standard for ultrastructural preservation and minimizing fixation-induced artifacts, as supported by extensive literature. This is why we adopted high-pressure freezing and freeze substitution in our study.

We have also published a separate methods paper (Tsang et al., eLife, 2018) directly comparing our cryofixation-based protocol with conventional chemical fixation, demonstrating substantial improvements in morphological preservation. This provides strong empirical support for the reliability of our approach.

Regarding the suggestion to validate observed morphological variation across populations: we note that determining the presence of artifacts requires a known ground truth, which is inherently unavailable as we could not measure the morphometrics of fly olfactory receptor neurons in their native state. In the absence of such a benchmark, we have instead prioritized using the best-available preparation methods and high-resolution imaging to ensure structural integrity.

Addressing these concerns and integrating additional experiments would significantly bolster the manuscript's completeness and advancement.

We appreciate the reviewer’s feedback. As discussed in our responses to the specific comments above, certain suggested experiments are currently limited by technical constraints, particularly in the context of high-resolution volume EM for insect tissues enclosed in cuticles.

Nevertheless, we have carefully addressed the reviewer’s concerns to the fullest extent possible within the scope of this study. We have revised the manuscript to clarify methodological limitations, added new explanatory content where appropriate, and ensured that our interpretations remain well grounded in the data. We hope these revisions strengthen the clarity and completeness of the manuscript.

**Reviewer #3 (Public Review):**
In the current manuscript entitled "Population-level morphological analysis of paired CO2- and odor-sensing olfactory neurons in *D. melanogaster* via volume electron microscopy", Choy, Charara et al. use volume electron microscopy and sensillum. They aim to investigate the degree of dendritic heterogeneity within a functional class of neurons using ab1Cand ab1D, which they can identify due to the unique feature of ab1 sensilla to house four neurons and the stereotypic location on the third antennal segment. This is a great use of volumetric electron imaging and neuron reconstruction to sample a population of neurons of the same type. Their data convincingly shows that there is dendritic heterogeneity in both investigated populations, and their sample size is sufficient to strongly support this observation. This data proposes that the phenomenon of dendritic heterogeneity is common in the *Drosophila* olfactory system and will stimulate future investigations into the developmental origin, functional implications, and potential adaptive advantage of this feature.Moreover, the authors discovered that there is a difference between CO2- and odour-sensing neurons of which the first show a characteristic flattened and sheet-like structure not observed in other sensory neurons sampled in this and previous studies. They hypothesize that this unique dendritic organization, which increases the surface area to volume ratio, might allow more efficient CO2 sensing by housing higher numbers of CO2 receptors. This is supported by previous attempts to express CO2 sensors in olfactory sensory neurons, which lack this dendritic morphology, resulting in lower CO2 sensitivity compared to endogenous neurons.Overall, this detailed morphological description of olfactory sensory neurons' dendrites convincingly shows heterogeneity in two neuron classes with potential functional impacts for odour sensing.Strength:The volumetric EM imaging and reconstruction approach offers unprecedented details in single cell morphology and compares dendrite heterogeneity across a great fraction of ab1 sensilla. The authors identify specific shapes for ab1C sensilla potentially linked to their unique function in CO2 sensing.

We thank the reviewer for the insightful comments and appreciation for our study.

Weaknesses:While the morphological description is highly detailed, no attempts are made to link this to odour sensitivity or other properties of the neurons. It would have been exciting to see how altered morphology impacts physiology in these olfactory sensory cells.

We agree that linking morphological variation to physiological properties, such as odor sensitivity, would be a highly valuable direction for future research. However, the aim of the current study is to provide an in-depth nanoscale characterization based on a substantial proportion of ab1 sensilla, highlighting morphological heterogeneity among homotypic ORNs.

At present, it is technically challenging to correlate the nanoscale morphology of individual ORNs with their physiological responses, as this would require volume EM imaging of the exact neurons recorded via single-sensillum electrophysiology. Currently, no dye-labeling method exists that is compatible with both single-sensillum recording and the stringent requirements of SBEM sample preparation to allow for unambiguous identification and segmentation of recorded ORNs.

To acknowledge this important limitation, we have added a paragraph in the *Discussion* section clarifying the current technical barriers and highlighting this as a promising area for future methodological development. Please also see our responses to the reviewer’s 4th comment below, where we present preliminary experiments examining whether odor sensitivity varies among homotypic ORNs.

(Please see the following pages for additional responses to the reviewers’ specific comments. These responses are not intended for publication.)

**Reviewer #1 (Recommendations for the authors):**
As this is mainly a descriptive paper I have no suggestions for additional experiments. Minor Text Suggestions:(1) The authors might want to include a better description/definition of the fly antennae, olfactory sensilla and their basic structure/makeup, position of the sensory neurons and dendrites within, etc, in the introduction perhaps in cartoon form to help readers that are not familiar (i.e. non-*Drosophila* readers) with the terminology and basic organization can follow the paper more easily from the start.

We thank the reviewer for the helpful suggestion to broaden the appeal of our study to a wider readership. In response, we added a new introductory paragraph at the beginning of the *Results* section, along with illustrations in a new supplementary figure (Figure 1—figure supplement 1). The new paragraph reads as follows.

“The primary olfactory organ in *Drosophila* is the antenna, which contains hundreds olfactory sensilla on the surface of its third segment (Figure 1—figure supplement 1A) . Each sensillum typically encapsulates the outer dendrites of two to four ORNs. The outer dendrites are the sites where odorant receptors are expressed, enabling the detection of volatile chemicals. A small portion of the outer dendrites lies beneath the base of the sensillum cuticle. At the ciliary constriction, the outer dendrites connect to the inner dendritic segment, which then links to the soma of each ORN (Figure 1—figure supplement 1B).”

(2) In Figure 4D, the letter annotations above the graphs are not clearly defined anywhere that I could easily find. Please clarify with different symbols and/or in the figure legend so readers can easily comprehend the stats that are presented.

We thank the reviewer for raising this point. As suggested, in the revised Figure 4D legend, following the original sentence “Statistical significance is determined by Kruskal-Wallis one-way ANOVA on ranks and denoted by different letters”, we added “For example, labels “a” and “b” indicate a significant difference between groups (P < 0.05), whereas labels with identical or shared letters (e.g., “a” and “a”, “a,b” and “a”, or “a,b” and “b”) indicate no significant difference.”

**Reviewer #3 (Recommendations for the authors):**
There are several aspects that I would like the authors to consider to improve the current manuscript:(1) Line 331: "Our analysis highlights how structural scaling in ab1D neurons achieves enhanced sensory capacity while maintaining the biophysical properties of dendrites". This is a strong statement, and not shown by the authors. They speculate about this in the discussion, but I would like them to soften the language here.

We thank the reviewer for raising this point. As suggested, we have softened the language in the sentence in question. The revised version is as follows.

“Our analysis suggests that structural scaling in ab1D neurons may enhance sensory capacity while preserving the biophysical properties of dendrites.”

(2) The Supplementary material is not well presented and is not cited in the manuscript. It is not clear what the individual data files show, where they refer to, etc. Please provide clear labels of all data, cite them at the appropriate location in the manuscript, and make them more accessible to the reader. Also, there are two Videos mentioned in the manuscript that are not included in the submission.

We thank the reviewer for bringing this to our attention and apologize for the oversight. We appreciate the reviewer’s careful attention to the supplementary materials. We have addressed these issues accordingly: (1) all source data have been consolidated in to a single, clearly labeled Excel file to improve accessibility for readers; this file is now cited at the appropriate locations in the manuscript. (2) The supplementary videos mentioned in the manuscript have also been included in the re-submission.

(3) In Figure 1B, it is hard to recapitulate the increase in dendritic density in the presented pictures. Could the authors please highlight dendrites in the raw imaging files (e.g. by colour coding as done later in the manuscript). Also, it might be helpful to indicate the measured parameters visually in this Figure (e.g. volume, length, etc.).

We thank the reviewer for the helpful suggestion. As suggested, we have pseudocolored the dendrites in Figure 1B to enhance visual clarity.

As noted, the original legend stated that “the sensilla were arranged from left to right in order of increasing dendritic branch counts”. To improve clarity, we have now added the number of dendritic branches above each sensillum to make this information more explicit.

We hope these changes make the figure more accessible and informative for readers.

(4) Given the strength of the authors in in vivo physiology and single sensilla recordings, I would be very curious about how the described morphological heterogeneity is reflected in the response properties of ab1Cs and ab1Ds. Can the authors provide data (already existing from their lab) of these two neurons on response heterogeneity? I acknowledge that spike sorting can be very challenging in ab1s, but maybe it is possible to show the range of response sensitivities upon CO2 stimulation in ab1Cs? The authors speculate in the discussion and presented data will only be correlative - however I think it would strengthen the manuscript to have some link to physiology included.

We thank the reviewer for this insightful comment. We share the same curiosity about response variability among homotypic ORNs, including ab1C and ab1D. Ideally, this question could be addressed by recording from a large proportion of neurons of a given ORN type to assess the response variability within a single antenna. However, due to technical limitations, we are only able to reliably record from 3–4 ab1 sensilla per antennal preparation, representing approximately 8% of the total ab1 population.

Moreover, our recordings are typically limited to ab1 sensilla located on the posterior-medial side of the antenna, as this region provides the best accessibility for our recording electrode. This spatial constraint may limit our ability to sample the full morphological diversity of ab1C and ab1D neurons.

Given these limitations, it is technically challenging to rigorously assess physiological variability in ab1C and ab1D responses across the entire ab1 population. Nonetheless, we attempted to address this question using a different sensillum type where a larger proportion of the population is accessible to single-sensillum recording per antennal preparation. Specifically, we focused on ab2 sensilla in the following analysis because we can reliably record from 6 sensilla per antenna, representing approximately 25% of the total ab2 population.

In the preliminary data presented below, we recorded from 6 ab2A ORNs per antenna across a total of 6 flies. Spike analysis revealed that odor-evoked responses were consistent across individual ab2A neurons (Author response image 1A). When analyzing the dose-response curve for each ORN, we found no statistically significant differences in odor sensitivity, either among ORNs within the same antenna or across different flies (Author response image 1B; two-way ANOVA: *P* > 0.99 within antennae, *P* > 0.99 across flies). This is further supported by the closely clustered EC50 values (Author response image 1C). This result suggests that odor sensitivity is largely uniform among homotypic ab2A ORNs.

**Author response image 1. sa4fig1:** Homotypic ab2A ORNs display similar odorant sensitivity. (A) Single-sensillum recording. Raster plots of ab2A/Or59b ORN spike responses. Six ab2A ORNs from the same antenna were recorded per fly. Odor stimulus: methyl acetate (10-6). (B) Dose-response relationships of peak spike responses, normalized to the maximum response of the ORN to facilitate comparison of odor sensitivity. Each curve represents responses from a single ab2A ORN fitted with the Hill equation (n=36 ab2 sensilla from 6 flies). Responses recorded from the same antenna are indicated by the same color. Statistical comparisons between different ab2A ORNs from the same antenna (P > 0.99) or across flies (P > 0.99) were performed by two-way ANOVA. (C) Quantification of individual pEC50 values from (B), defined as -logEC50.

However, we are hesitant to include this result in the main manuscript for several reasons. First, it does not directly relate to the morphometric analysis of ab1C and ab1D neurons, which is the primary focus of our study. Second, while we were able to record from approximately 25% of the ab2 population, this level of coverage is still limited and potentially subject to sampling bias due to the spatial constraints of the antennal region accessible to the recording electrode.

At best, our data suggest limited variability in odor sensitivity among the recorded ab2A ORNs. However, we are cautious about generalizing this finding to the entire ab2 population. In light of these considerations, we hope the reviewer can appreciate the technical challenges inherent in addressing what may appear to be a straightforward question.

For these reasons, we have chosen to include this preliminary result in the response only, rather than in the main manuscript.